# Simple statistics for complex Earthquakes' time distribution

Teimuraz Matcharashvili[1], Takahiro Hatano[2], Tamaz Chelidze[1], and Natalia Zhukova[1]

[1]M. Nodia Institute of Geophysics, Tbilisi State University, Tbilisi, Georgia
[2]Earthquake Research Institute, the University of Tokyo, Tokyo, Japan

*Correspondence to:* T.Matcharashvili (matcharashvili@gtu.ge)

**Abstract.** Here we investigated a statistical feature of earthquakes time distribution in southern Californian earthquake catalogue. As a main data analysis tool, we used simple statistical approach based on the calculation of integral deviation times (IDT) from the time distribution of regular markers. The research objective is to define whether and when the process of earthquakes time distribution approaches to randomness. Effectiveness of the IDT calculation method was tested on the set of simulated color noise data sets with the different extent of regularity as well as for Poisson process data sets. Standard methods of complex data analysis have also been used, such as power spectrum regression, Lempel and Ziv complexity and recurrence quantification analysis as well as multi-scale entropy calculation. After testing the IDT calculation method for simulated model data sets, we have analyzed the variation of the extent of regularity in southern Californian earthquake catalogue. Analysis was carried out for different periods and at different magnitude thresholds. It was found that the extent of the order in earthquakes time distribution is fluctuating over the catalogue. Particularly, we show that the process of earthquakes' time distribution becomes most random-like in periods of relatively decreased local seismic activity.

*Copyright statement.* TEXT

## 1 Introduction

Time distribution of earthquakes remain one of the important questions in nowadays geophysics. At present, the results of theoretical research and the analysis of features of earthquakes' temporal distributions from different seismic regions with different tectonic regimes carried worldwide can be found in (e.g., Matcharashvili et al., 2000; Telesca, 2001, 2012; Corral, 2004; Davidsen and Goltz, 2004; Martinez et al., 2005; Lennartz et al., 2008; Chelidze and Matcharashvili, 2007).

Such analyses among others often aims to the assessment of the strength of correlations or the extent of the determinism/regularity in the earthquakes time distribution. One of the main conclusions of such analysis is the understanding that earthquake generation in general does not follow the patterns of random process. Exactly, well established clustering, at least in time (and spatial domains), suggests that earthquakes are not independent completely and that seismicity is characterized by slowly decaying correlations (named long-range correlations): such behavior is commonly exhibited by non-linear dynamical systems far from equilibrium (Peng et al., 1994, 1995). Moreover, it was shown that in the temporal and spatial domains earthquakes' distribution may reveal some features of a low-dimensional, nonlinear structure, while in the energy domain (mag-

nitude distribution) it is close to a random-like high dimensional process (Goltz, 1998; Matcharashvili et al., 2000). Moreover, according to present views, the extent of regularity of the seismic process should vary in time and space (Goltz, 1998; Matcharashvili et al., 2000, 2002; Abe and Suzuki, 2004; Chelidze and Matcharashvili, 2007; Iliopoulos et al., 2012).

At the same time, the details of how the extent of randomness (or non-randomness) of seismic process changes over the time and space still remain unclear. In the present work, on the basis of southern California earthquake catalogue, we aimed to be focused on this question and analyzed earthquake time distribution to find where it is closer to randomness.

## 2 Data and used Methods

Our analysis is based on the southern Californian earthquakes catalogue available from http://www.data.scec.org/ftp/catalogs/. As far as we aimed to analyze temporal features of original earthquakes generation, we tried to have as long as possible period of observation with as low as possible representative threshold. For this purpose, according to results of time completeness analysis (not shown here) we decided to be focused on the time period from 1975 to 2017 since in the middle of $70^{th}$ of last century Mc clearly decreased. Southern Californian (SC) catalogue for the considered period is complete for M = 2.6, according to the Gutenberg–Richter relationship analysis.

In general, presently we are developing an approach aiming to discern features of the complex data sets (in this case EQ time distribution) by comparing them with data sets with the predefined dynamical structures. In the present work in the frame of this general idea we started from the simplest case, comparing natural time distribution of earthquakes in SC catalogue with the time distribution of regular markers, according to the scheme shown in Fig. 1.

Namely, knowing duration of whole period of observation in considered catalogue (22167178 minutes, from 01.01.1975 to 23.02. 2017) and the number of earthquakes (34020) with the magnitude above a representative threshold (M2.6), we calculated the time step between consecutive regular markers (651.6 min),what in fact is the mean time of earthquakes occurrence for the considered period. Then, for each of earthquakes in the catalogue we calculated difference between original event occurrence time and time point of the regular marker. We denoted as $DT(i)$ the time interval (delay or deviation time) between occurrence of original earthquake $T_{EQ(i)}$ and corresponding $i - th$ regular marker $T_{R(i)}$. It is clear that original earthquake $(EQ_i)$ may occur prior or after of corresponding regular marker $(R_i)$, so by $DT_{(i)}$ with minus or plus sign we understand earthquakes occurred prior or after regular markers accordingly. Summation of deviation times may provide interesting knowledge about character of distribution of earthquakes comparing to regular time markers. Here we mention that, alternatively, the same can be viewed as a summation of differences (deviations) between observed waiting times and mean occurrence time over considered period.

In any case,logically, for any random sequence, the sum of the deviation times should approach zero when $n \to \infty$. Thus, the main assumption is that the integral of deviation times (IDT) or the sum $\sum_{i=1}^{N} DT_{(i)} \to 0$ when the time distribution of events is random-like. From this point of view the used approach looks close to Cumulative Sums (Cusum) test, where for a random sequence, the sum of excursions of the random walk should be near zero (Rukhin et.al., 2010).

Prior to use for the seismic process, we needed to test whether IDT calculation can be sensitive to dynamical changes occurred

in complex data sets with known dynamical structures. We started from the analysis of model data sets with a different extent of randomness. Exactly, we used simulated noise data sets of different color with power spectrum function $(1/f^\beta)$, where scale exponent $(\beta)$ varied from about 0 to 2. These noises, according to generation principles, logically have to be different, but for purposes of our analysis we needed to have strong quantitative assessments of such differences. This is why at first, these noise

data sets have been investigated by several data analysis methods, often used to assess different aspects of changes occurred in dynamical process of interest. Exactly, power spectrum regression, Lempel and Ziv algorithmic complexity calculation as well as recurrence quantification analysis and Multiscale entropy calculation method have been used for simulated model data sets. All these popular methods of time series analysis are well described in number of research articles and we will just briefly mention their main principles.

Power spectrum regression exponent calculation enables to elucidate scaling features of data set in the frequency domain. By this method a fractal property is reflected as a power law dependence between the spectral power $(S(f))$ and the frequency $(f)$ by spectral exponent $\beta$:

$$S(f) \sim \frac{1}{f^\beta} \tag{1}$$

$\beta$ often is regarded as a measure of the strength of the persistence or anti-persistence in data set. As easily calculated from log-log power spectrum plot, $\beta$ is related to the type of correlations present in time series (Malamud, 1999; Munoz-Diosdado et al., 2005; Stadnitski, 2012). For example, $\beta = 0$, corresponds to the uncorrelated white noise, and processes with some extent of memory or long-range correlations are characterized by nonzero values of spectral exponents.

Next, we proceeded to the Lempel and Ziv algorithmic complexity (LZC) calculation (Lempel and Ziv, 1976; Aboy et al.,

2006; Hu and Gao, 2006), which is a common method for quantification of the extent of order (or randomness) in data sets of different origin. LZC is based on the transformation of analyzed sequence into new symbolic sequence. For this original data are converted into a 0, 1, sequence by comparing to certain threshold value (usually median of the original data set). Once the symbolic sequence is obtained, it is parsed to obtain distinct words, and the words are encoded. Denoting the length of the encoded sequence for those words, the LZ complexity can be defined as

$$C_{LZ} = \frac{L(n)}{n} \tag{2}$$

where $L(n)$ is the length of the encoded sequence and $n$ is the total length of sequence (Hu and Gao, 2006). Parsing methods can be different (Cover, 1991; Hu and Gao, 2006). In this work, we used scheme described in (Hu and Gao, 2006).

Next, in order to further quantify changes in the dynamical structure of simulated data sets, we have used recurrence quantification analysis (RQA) approach (Zbilut and Webber, 1992; Webber and Zbilut, 1994; Marwan et al., 2007). RQA is often used

for analysis of different types of data sets and represents a quantitative extension of Recurrent Plot (RP) construction method. RP, itself, is based on the fact that returns (recurrence) to the certain condition of the system (or state space location) is a fundamental property of any dynamical system with quantifiable extent of determinism in underlying laws (Eckman, 1987). In

order to successfully fulfill RQA calculations, at first the phase space trajectory should be reconstructed from the given scalar data sets. It is important to test the proximity of points of the phase trajectory by the condition that the distance between them is less than a specified threshold $\varepsilon$ (Eckman, 1987). In this way, we obtain two-dimensional representation of the recurrence features of dynamics,which is embedded in a high-dimensional phase space. Then a small-scale structure of recurrence plots
can be quantified (Zbilut and Webber, 1992; Webber and Zbilut, 1994, 2005; Marwan et al., 2007; Webber et al., 2009; Webber and Marwan, 2015). Namely, RQA method enables to to quantify features of a distance matrix of recurrence plot, by means of several measures of complexity. These measures of complexity are based on the quantification of diagonally and vertically oriented lines in the recurrence plot. In this research we calculated one of such measures: the percent determinism ($\%DET$) which is defined as the fraction of recurrence points that form diagonal lines of recurrence plots and which shows changes in
the extent of determinism in the analyzed data sets.

An additional test, which we used to quantify the extent of regularity in modeled data sets, is the composite multi-scale entropy (CMSE) calculation (Wu et al., 2013a). CMSE method represents expansion of multi-scale entropy (MSE) [Costa, et al. 2005] method, which in turn originates from the concept of sample entropy (SampEn) (Richman and Moorman, 2000). SampEn is regarded as an estimator of complexity of data sets for a single time scale. In order to capture the long-term structures in the time
series, further Costa (2015) proposed the above mentioned multiscale entropy (MSE) algorithm, which uses sample entropies (SampEn) of a time series at multiple scales. At the first step of this algorithm, often used in different fields, a coarse-graining procedure is used to derive the representations of a system's dynamics at different time scales; at the next step, the SampEn algorithm is used to quantify the regularity of a coarse-grained time series at each time scale factor. Main problem of MSE is that, for a shorter time series, the variance of the entropy estimator grows very fast as the number of data points is reduced.
In order to avoid this problem and reduce the variance of estimated entropy values at large scales, a method of the composite multi-scale entropy (CMSE) calculation was developed by Wu and colleagues (Wu et al., 2013a).

## 3   Results and discussion

### 3.1   Analysis of model data sets

As we mentioned in previous section, first, we needed to ascertain whether calculation of IDT values is sensitive to dynamical
changes, occurred in analyzed data sets. To this end, we decided to generate artificial datasets of one and the same type, for example noises, which according to the generation procedure should be measurably different in the frequency content, representing a different types of color noises. We have started from the analysis of 34020 data length sequences of these noise data sets. For clarity we add here that to test the robustness of results, the same analyses were performed on much longer data sets, but here we show results for simulated noise data sets, which are of the same length as the original data sets from the
used seismic catalogue. The noise data sets have been generated according to concepts described in Kasdin (1995), Milotti (2007) and Beran et al. (2013). As a metrics for these data sets we have used mentioned above power spectrum exponents ($\beta$), also referred to as the spectral indexes (Schaefer et al., 2014). Exactly, we have analyzed seven of such data sets having spectral exponents in the range: 0.001, 0.275, 0.545, 0.810, 1.120, 1.387, 1.655. Values of $\beta$, are often used as a metric for

the fractal characteristics of data sequences (Shlesinger, 1987; Schaefer et al., 2014). In our case different spectrum exponents of simulated noise data sets indicate that they are different by the extent of correlations in the frequency content (Schaefer et al., 2014). Indeed, the first noise set, with the $\beta = 0.001$ (Fig.2, a), was closest to the white noise and the last one, with the $\beta = 1.655$ (Fig.2, b), manifested the features closer to colored noises of red or Brownian type, with detectable dynamical

structure. In addition to this, taking into account that we aimed to analyze seismic data sets, we regarded as logical to consider also the random process, which is often used by seismologists – a Poisson process. We generated the set of 34020 data long sequences of Poisson process. It was quite expectable that spectral exponent of such sequences is close to that of a white noise. For further analysis, in order to differentiate simulated (noise and Poisson process) data sets by the extent of randomness, we used algorithmic complexity (LZC) and recurrence quantification analysis methods as well as testing based on multi-scale

entropy (MSE) analysis.

In Fig. 3, we show results of LZC and $\%DET$ calculation; namely here are presented averages of values calculated for consecutive 1000 data windows shifted by 100 data. Both methods, though based on different principles, help to answer question, how similar or dissimilar are considered data sets by the extent of randomness. We see that, Lempel and Ziv complexity measure decreases from 0.98 to 0.21, when $\beta$ of noises increases. It means that the extent of regularity in simulated data sets increases.

The same conclusion is drawn from RQA: the percentage of determinism increases from 25 to 96.5 when the spectral exponent increases.For both LZC and RQA measures, differences of compared neighbor groups in figures are statistically significant at p=0.01. Thus, according to Fig. 3, the extent of regularity in simulated noise sequences clearly increases from the close to white ($\beta = 0.001$) to close to Brownian ($\beta = 1.655$) noise. For the Poisson process data sequences, LZC measure reaches 0.97-0.98 and $\%DET$ is in the range 25-26, i.e. these values are close to what we obtain for white noises.

Thus, the results of LZC and $\%DET$ calculations confirm the result of power spectrum exponents calculations, show that considered color noise data sets are different from white noise and Poisson process by the extent of regularity.

Additional multi-scale, CMSE, analysis also shows (Fig. 4) that the extent of regularity in model noise data sets increases, when they become "more" colored (from $\beta = 0.001$ to $\beta = 1.655$). We see that for small scales (exactly for scale one and partly scale two), noise data sets reveal decrease in the entropy values for simulated data sets, when spectral indexes rise from

$\beta = 0.001$ to $\beta = 1.655$. This is logical for simulated data sets, where the extent of order, according to the above analysis, should slightly increase. At the same time, while at larger scales, the value of entropy for the noise data set with $\beta = 0.001$ continue monotonically decreases like for coarse-grained white noise time series (Costa, 2015). On the other hand the value of entropy for 1/f type processes with the $\beta$ values close to pink noises (0.81, 1.12) remained almost constant for all scales. As noticed by Costa (2015), this fact was confirmed in different articles on multi-scale entropy calculation [see e.g. (Chou, 2012;

Wu et al., 2013a, b)]. Costa and coauthors explained this result by the presence of complex structures across multiple scales for 1/ f type of noises. From this point of view, in color noise set, closer to a Brownian type process, the emerging complex dynamical structures should become more and more organized. Apparently, these structures are preserved over multiple scales including small ones. This is clearly indicated by the gradual decrease of calculated values of entropy for sequences with $\beta = 1.12$ to $\beta = 1.387$ and $\beta = 1.654$ at all considered scales (see Fig. 4).Poisson process data sets (not shown in figure) in the

sense of results of multiscale analysis are close to a white noise sequence with $\beta = 0.001$.

Thus, CMSE analysis additionally confirms that used in this research complex model data sets are characterized by quantifiable dynamical differences. Once we had data sets with the quantifiable differences in their dynamical structures, we started to test the ability of IDT calculation to detect these differences.

For this, we created cumulative sum sequences of seven noise data sets and data sets of Poisson process and regarded them as models of time occurrences of consecutive events. We treated these, 34020 data long, sets for time occurrence sequence of real earthquakes and calculated IDT values for different windows. Taking into consideration that cumulative sum (or time span in the case of seismic catalogue) of windows may be different (excluding the case when data sets have been specially generated so that cumulative sum to be equal) we normed obtained IDT values to the span of window. Results of calculation are presented in the upper curve (circles) in Fig. 5a. Here also we present results of similar calculations on the same data sets performed for shorter windows (see squares, triangles and diamonds in Fig. 5a).

As we see, absolute values of normed to window span IDTs, calculated for the model data sets indicate stronger deviation from zero, when the extent of order in simulated noise data sets increases (according to the above analysis). Average values of IDTs calculated for data sets with spectral exponents closer to Brownian noise significantly differ from white noise at p=0.01 (Fig. 5 a). It needs to be pointed that comparing to results obtained by used above methods, IDT calculation looks even more sensitive to slight dynamical changes occurred in simulated data sets; note more than 1.5 order difference between sequences with $\beta = 0.001$ and $\beta = 1.654$ for the entire length of data sets (circles in Fig.5a).

It needs to be pointed, that according to IDT calculations, Poisson process looks more random than white noise. Indeed, logarithms of normed to window span IDT values calculated for random Poisson process data sets were lower than for white noise (0.38, 0.1, 0.04, 0.03 for 30000, 20000, 10000 and 5000 data sets accordingly). For the further analysis, it is important to mention that results of above calculations do not practically depend on the length of used data sets. Not the less important is that, as it is shown in Fig. 5b, differences found for longer windows is preserved for the short, 100 data long sequences. For 100 data windows difference between white noise, as well as Poisson process, data sets and colored noises is statistically significant at p=0.01. Taking into consideration the importance of results of IDT calculations for short (100 data) windows, we additionally present reconstructed PDF curves fitted to the normal distribution according to real calculations (different marks in Fig. 6). From this figure we see that IDT values goes closer to zero when the extent of order decreases. Besides, it also becomes clear that even for short data sets IDT calculation is useful to detect differences in considered data sets.

Thus, based on the analysis of specially simulated data sequences we conclude that IDT calculation method is effective in detecting small changes occurred in, even short, complex data sets with different dynamical structures.

### 3.2    Analysis of earthquakes time distribution in south California catalogue

In this section we proceeded to the analysis of original data sets drawn from the south California seismic catalogue using IDT calculation approach.

As it was said above, for any random sequence, the sum of the deviation times should approach zero in the infinite length limit. Results of presented in the previous section analysis confirms this on the example of model random (or random-like) data sets with different extent of regularity (or randomness).

In the case of real earthquake generation process, which according to present views can not be regarded as completely random Goltz (1998); Matcharashvili et al. (2000, 2016); Abe and Suzuki (2004); Iliopoulos et al. (2012), we should assume that the integral of deviations times (IDTs) for the periods with the more random-like earthquake's time distribution will be closer to zero, compared to the less random ones.

To show this, we used seismic catalogue of south California, the most trustworthy data base for analysis like targeted in this research. Aiming at the analysis of temporal features of seismic process, we intentionally avoided any cleaning or filtering of catalogue in order to preserve its original temporal structure. Therefore, according to common practice [see e.g. Christensen et al. (2002); Corral (2004)] we regard the seismic processes in this catalogue as a whole, irrespective of the details of tectonic features, earthquakes location or their classification as mainshocks or aftershocks.

It was quite understandable that, for such catalogue we logically should expect time clustering of interdependent events: fore- and aftershocks. This, in the context of our analysis, apparently could lead to the considerable amount of events occurred prior to corresponding regular markers (probably mostly aftershocks). Thus, it was interesting to know how the number of events occurred prior or after regular markers and especially result of IDT calculation (which directly depends on the number of events occurred prior and after regular markers), is related with the time locations of such interdependent events.

Here we underline the fact that both IDT values as well as the portion of events occurred prior or after regular markers (as defined in the methods section) would strongly depend on the position and length of certain time window in catalogue.Thus, we focused on the whole duration period of considered catalogue (at the representative threshold M2.6). We found that in this catalogue 55% of all earthquakes occurred prior and 45% after the regular time markers. To elucidate the role of dependent events on this ratio we analyzed catalogue for higher representative thresholds. At increased to M3.6 representative magnitude

threshold we found that the portion of earthquakes occurred prior to markers decrease (33% of all earthquakes). This provided argument in favor of assumption that the prevalence of earthquakes, which occur prior to markers may indeed be related with low-magnitude dependent events (supposedly mostly aftershocks). At the same time, further increase of representative threshold convinces that low magnitude dependent events in catalogue may not be the sole cause influencing the amount of earthquakes occurred prior to markers. Indeed, the portion of events occurred prior to markers increased to 42% at the rep-

resentative threshold M4.6. Most noticeable is that at highest considered representative threshold, M5.6, we observe further increase of portion of earthquakes, occurred prior to regular markers; to the level observed for M2.6 threshold (55% of all events). Thus, it seems unlike that ratio between events that occurred prior or after regular markers may be related only with dependent events (aftershocks).

Next, we calculated IDT values for entire observation period at different representative thresholds. It was found that IDT value

calculated for the entire observation period of considered southern Californian earthquake catalog (at representative threshold M2.6), equals: -14611458375 minutes (as mentioned above sign 'minus' here denotes the direction of summary deviation along time axis). We compare this value to the IDT values at larger representative thresholds. Taking into account that increase of threshold may somehow change the analyzed period of catalogue, below we show normed to the corresponding time span of catalogue values of IDT. Namely, for M2.6 catalogue the normed value of IDT= -659.15. Increasing the threshold to M3.6,

M4.6, and M5.6 leads to following IDT values: 71.7, 6.7, -0.87 accordingly. Two important things can be underlined here:

first, the increase of the magnitude threshold makes the time distribution of remained EQs more random and second, according to our conjecture to the more random EQ distribution should corresponds the closer to zero IDT value, what indeed is shown above.

Interesting fact is that decreased probability of dependent events, at increased representative threshold, do not necessarily causes proportional increase of the number of occurred after regular markers events, though absolute values of IDT drastically decreases. These results also indicate that the ratio between events, occurred prior or after regular markers, found for the entire span of SC catalogue, as well as the IDT value, should not be reduced due only to the distributional features of dependent events.

Further we needed to clear up whether the ratio of events occurred prior or after regular markers and especially obtained IDT value, are characteristics of time distribution of earthquakes in the SC catalogue or they reflect influence of some unknown random effects, which are not directly related with the seismic process. For this we started to calculate IDT values for the set of randomized catalogues. In these artificial catalogues the original time structure of the southern Californian earthquakes distribution was preliminary destroyed. Exactly, occurrence times of original events' have been randomly shuffled (i.e. earthquakes' time locations have been randomly changed over the entire time span of more than 42 years). We have generated 150 of such randomized catalogues and for each of them, IDT values have been calculated for the whole catalogue time span (which was the same as for the original catalogue).

It was found that for the whole period of observation, in 58% of all time-randomized catalogues prevailed earthquakes, occurred prior to corresponding regular markers. At the same time, unlike the original catalogue, where 55% of earthquakes occurred prior to corresponding regular markers, in randomized catalogues the portion of such earthquakes, occurred prior to markers, varied in the wide range (from 51% to 92%). Thus, in spite of some similarity, by the portion of events occurred prior and after of regular markers original and time randomized catalogues are still different.

Next, comparing the averaged IDT value of randomized catalogues (calculated from IDTs of 150 randomly shuffled catalogues) we found that it is also with minus sign (-159755608 min). This was expectable since in 58% of cases of randomized catalogues, prevailed earthquakes occurred prior to regular markers. Thus, comparing the average of integral deviation times, calculated for the entire length of randomized catalogues, with the IDT value of the original SC catalogue, we see that the last one is two orders of magnitude larger. The difference between IDT of the whole original catalogue and the average IDT of randomized catalogues was statistically significant, with Z score =11.2, corresponding to p=0.001 Bevington and Robinson (2002); Sales-Pardo et.al. (2007).

The difference between IDT values calculated for original and time randomized catalogues is further highlighted in Fig. 7, where normed to the windows span IDT values are presented. We see, that in all cases normed to windows span IDTs are clearly smaller than for the original catalogue (6.59E+02 ). It is interesting that in at least 30% of cases IDTs calculated for randomized catalogues are more than two order smaller than IDT for original catalogue.

From this analysis two important conclusions can be drawn: i) IDT value, calculated for the considered period of south Californian earthquake catalogue, expresses the internal time distribution features of the original seismic process, and $ii$) random-like earthquake time distribution lead to lower (closer to zero) IDT values comparing to the whole original catalogue.

All above results obtained for simulated data sets as well as for the time distribution of earthquakes in the original catalogue shows undoubtedly that the time distribution of earthquakes in south California for the entire considered period should be regarded as a strongly non-random process (IDT is larger than for randomly distributed in time earthquakes). Therefore, result of this simple statistical analysis is in complete agreement with our earlier results, obtained by contemporary nonlinear data

analysis methods, as well as with the results of similar analysis reported by other authors, which used different methodological approaches see e.g.Goltz (1998); Matcharashvili et al. (2000, 2016); Abe and Suzuki (2004); Telesca (2012); Iliopoulos et al. (2012).

Thus, we found that for the whole period of considered catalogue, prevailed earthquakes occurred prior to corresponding regular markers (see also the last point in the upper curve of Fig. 8b). At the same time, as also was mentioned, the number of

earthquakes occurred prior or after corresponding regular markers may change depending on the time span of analyzed catalogue. The same can be said about the values of the integral deviations times. In order to investigate the character of the time variation of IDT values of South California earthquake catalogue in different periods we fulfilled calculation for the expanding time windows. Exactly, we have calculated IDT values starting from the first 100 data (earthquakes), expanding initial window by the consecutive 10 data to the end of catalogue. In Fig. 8, we see that the number and the time location of earthquakes

(relative to regular markers), undergoes essential change, when the length of analyzed part of catalogue (analyzed window's length) gradually expands to the end of catalogue (in our case from 01.01.1975 to 23.02. 2017).

As it is shown in Fig. 8a, in the most of the analyzed windows the majority of earthquakes occurred after regular markers, although there are windows with opposite behavior . So far as in the most windows prevail earthquakes which occurred after regular markers, it is not surprising that calculated for consecutive windows integral deviation times are mostly positive. This

is clear from Fig. 8b, (upper curve), where we see windows with negative IDTs too. Thus, the values of IDTs, calculated for extended windows in different periods vary in a wide range, increasing or decreasing and sometimes coming close to zero.

Here we point again, that based on results of above analysis, accomplished for simulated data sets and randomized catalogues, we suppose that when IDT value approaches zero, the dynamical features of originally nonrandom seismic process undergoes qualitative changes and becomes random-like or at least is closer to randomness. In other cases, when IDT value changes over

time, but is far from zero, we observe quantitative changes in the extent of regularity of nonrandom earthquakes time distribution.

From this point of view it is interesting that earthquakes' time distribution looks more random-like for the relatively quiet periods, when the amount of seismic energy calculated according to Kanamori (1997), decreases comparing to values, released in the neighbor windows prior or after strongest earthquakes. This is noticeable, in the lower curve of Fig. 8b, where we present

cumulative values of seismic energy, calculated for consecutive windows, expanding by 10 events to the end of catalogue. We see that in south California, from 1975 to 2017, the strongest earthquakes never occurred in periods, when IDT curve comes close to zero value or crosses abscissa line. To avoid misunderstanding because of restricted visibility in Fig. 8b, we point here that M6.4 earthquake has occurred in the window 256, from the start of catalogue, and IDT curve crossed abscissa later, in the window 265, from the start of catalogue, i.e. 100 events later.

Results in Fig. 8, also provides interesting knowledge about relation between IDT and the amount of released seismic energy.

As we see, three strongest earthquakes in southern Californian earthquake catalogue (1975-2017) occurred on the rising branch of IDT curve close or immediately after local minima.This local decrease of IDT values, possibly, points to the decreased extent of regularity (or increased randomness) in the earthquakes temporal distribution in periods prior to strongest earthquakes in California.

In order to avoid doubts related to the fixed starting point of the above analysis, we have carried out the same calculation of IDT values for catalogues, which started in 1985, 1990, 1995, 2000 and 2005. As we see in Fig. 9, analysis carried out on shorter catalogues confirm the result obtained for the entire period of observation (1975-2017) and convinces that the curve of IDT values crosses abscissa at periods of relatively decreased seismic energy release.The case of M6.4 earthquake in Fig. 8b, is not an exception, as we explained above.

For better visibility of changes in the process of energy release, in Fig. 9 (bottom) we show increments of seismic energy release calculated for only the last 10 events in each consecutive windows, opposite to Fig. 8, where we presented energies released by all earthquakes in each window. This was done to make more visible the fine structure of changes in energy release in the expanding (by consecutive 10 events) part of windows, which otherwise is hidden by the strong background level of the summary energy release in the whole window. At the same time, we should not forget that IDTs in Fig.9 are calculated for the

entire length of windows and that real evolution of energy release looks similar to presented in Fig.8 b.

Thus we see that shortening the time span of the analyzed part of catalogue does not influence obtained results.

We above already discussed influence of increased representative threshold on the calculated for entire catalogue span IDT value. Now, it was necessary to check, how the change of representative threshold will influence obtained results for expanding windows. This was a very important aspect of our analysis, because there is a well known point of view that the time distribu-

tion of large (considered as independent events - coupling between which is exception rather than a rule) and medium-small earthquakes (for which time distribution may be governed or triggered by the interaction between events) is significantly different (Lombardi and Marzocchi, 2007).

To see how the results of integral deviation times analysis may be influenced by considering smaller or stronger earthquakes we carried out analysis of south California catalogue for earthquakes above M3.6 and M4.6 thresholds. Analysis (see results in

Figs. 10 and 11) has been accomplished in a manner, similar to the scheme for threshold M2.6, i.e. for the entire available period 1975-2017 and for shorter periods (from 1985, 1990, 1995, 2000, 2005 to 2017). Further analysis by the same scheme for higher (e.g. M5.6) threshold magnitudes was impossible because of the scarce number of large earthquakes in the considered seismic catalogue (just 29 earthquakes above M5.6). At the same time we point out that even for M5.6 representative threshold, for the entire period 1975-2017, the results obtained for two or three available windows (29 events at windows expanding by 9

or 10 data) agree with the above results showing that a lower IDT value corresponds to period with decreased seismic energy release.

Thus, we conclude that the increase of magnitude threshold (Figs. 10 and 11) practically do not change the results found for lower representative threshold. This means that increasing representative threshold we still deal with the catalogue in which relatively small and medium size events prevail. Therefore, conclusions drawn from the analysis for original representative

threshold (M2.6) remain correct for the case, when we consider a catalogue with relatively stronger events; thus it seems that

there is no principal difference in the character of the contribution of smaller and stronger events to the results of IDT calculation. Comparison with the results obtained for time randomized catalogues confirm this conclusion.

Because of mentioned above unclearness in Figs. 9-11, when we calculated IDTs for the expanding windows and discuss results for the energy release occurred in the last 10 data windows, we accomplished additional analysis on the sliding windows with fixed number of events. In detail, in the South Californian earthquake catalogue we have calculated IDT values for non-overlapping windows of 100 consecutive events, shifted by 100 data (Figs. 12, 13). We have used short sliding windows of 100 data for two reasons: i. to have good resolution of changes occurred in the time distribution of earthquakes and because, ii. even relatively short, 100 data span windows also provide good enough discrimination in the IDT values, as it is shown in Figs. 5b and 6.

In Fig. 12 (top), we see that for the entire period of analysis there are dozens of IDT values that are not far from one tenth of the standard deviation ($\sigma$) from zero (given by black circles in the top figure). Most importantly, among them 8 of IDT values are within of $0.01\sigma$ to zero. These values of IDT, (shown in black in the middle figure), can be regarded practically equal to zero. According to the above results on simulated and original data sets, seismic process in the windows with close to zero values of IDT can be regarded as random. If we compare occurrence of these practically zero IDT values with the amount of seismic energy released in consecutive windows (bottom in Fig. 12) it becomes clear that they occur in periods of essentially decreased (comparing to observed maximums) seismic energy release. Similar conclusion is drawn from the analysis of catalogues for earthquakes above M3.6 threshold (Fig.13). Because of the restricted number of strong events in the catalogue, further increase of the threshold magnitude was impossible for the case of 100 data non-overlapping sliding windows.

Results obtained for non-overlapping sliding windows of fixed length also confirm the results obtained for expanding windows. Simple statistical approach used here thus shows that extent of randomness in the earthquakes time distribution is changing over time and that it is most random-like at periods of decreased seismic activity. The results of this analysis provide additional indirect arguments in favor of our earlier suggestion that the extent of regularity in the earthquakes time distribution should decrease in seismically quiet periods and increase in the periods of strong earthquakes preparation (Matcharashvili et al., 2011, 2013).

## 4 Conclusions

We investigated earthquakes time distribution in the Southern Californian earthquake catalogue by the method of calculation of integral deviation times relative to regular time marks. The main goal of research was to quantify, when the time distribution of earthquakes become closer to the random process. Together with IDT calculation, standard methods of complex data analysis such as power spectrum regression, Lempel and Ziv complexity and recurrence quantification analysis, as well as multi-scale entropy calculation have also been used. Analysis was accomplished for different time intervals and for different magnitude thresholds. Based on a simple statistical analysis results, we infer that the temporal distribution of earthquakes in Southern Californian catalogue is the most random-like at the periods of decreased local seismic activity.

*Competing interests.* The authors declare that they have no conflict of interest." This statement should come prior to the acknowledgements.

*Acknowledgements.* This work was supported by Shota Rustaveli National Science Foundation (SRNSF), grant 217838 "Investigation of dynamics of earthquake's temporal distribution".

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

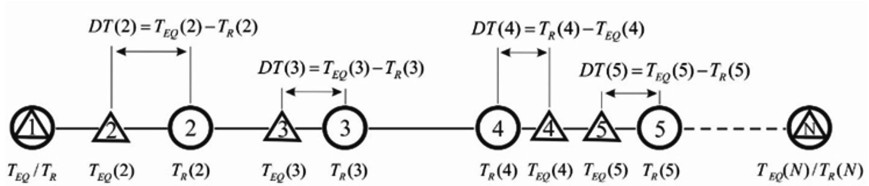

**Figure 1.** Explanation of the used approach. Triangles - time location of original earthquakes ($T_{EQ}(i)$), circles – time location of regular markers ($T_R(i)$). $DT_{(i)}$ denotes the difference between the time of earthquake occurrence ($T_{EQ}(i)$) in the catalogue and the time point of the regular marker ($T_R(i)$) .

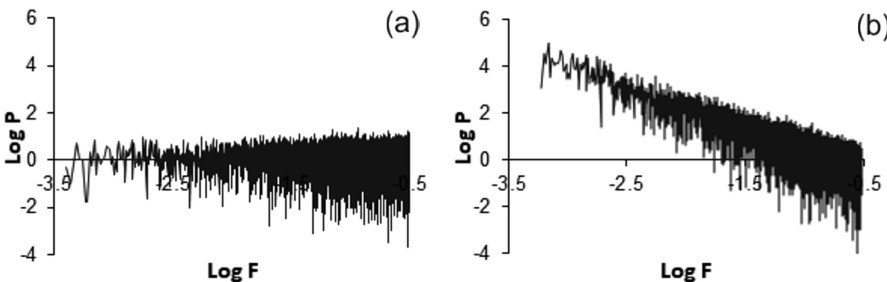

**Figure 2.** Typical plot of the power spectrum of simulated data sets with different spectral regression, a)$\beta = 0.001$ and b)$\beta = 1.655$

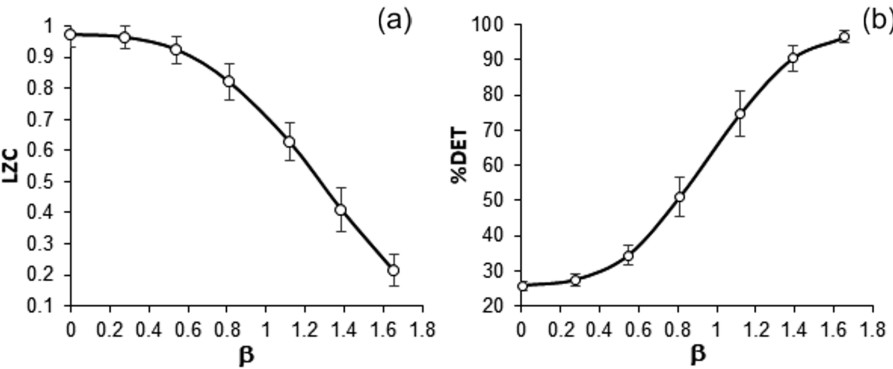

**Figure 3.** LZC and $\%DET$ values calculated for seven noise data sets with different spectral indexes.

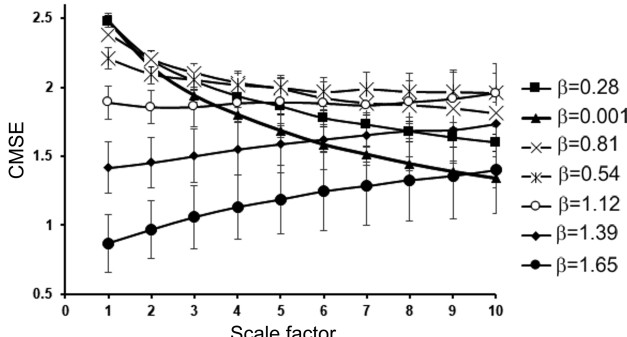

**Figure 4.** CMSE values versus scale factor for simulated data sequences with different spectral indexes.

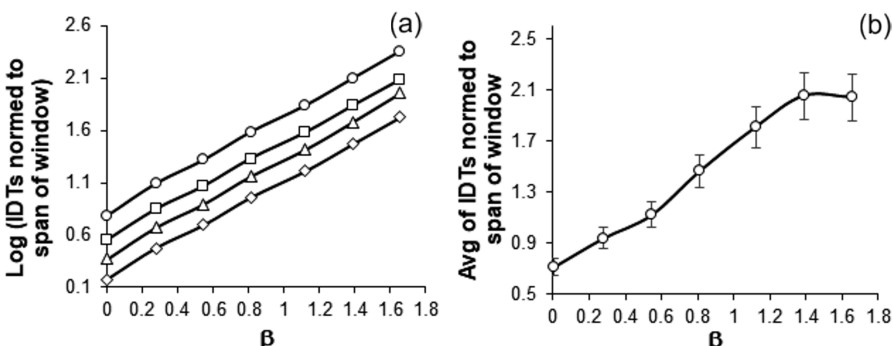

**Figure 5.** a) Logarithms of, normed to the span of window, absolute values of IDT calculated for different length (circles-34020, squares-20000, triangles-10000, diamonds-5000 data) of windows of simulated noise data sets with different spectral indexes, b) averages of IDT values calculated for 100 data windows of normed to the span of window simulated noise data sets with different spectral indexes.

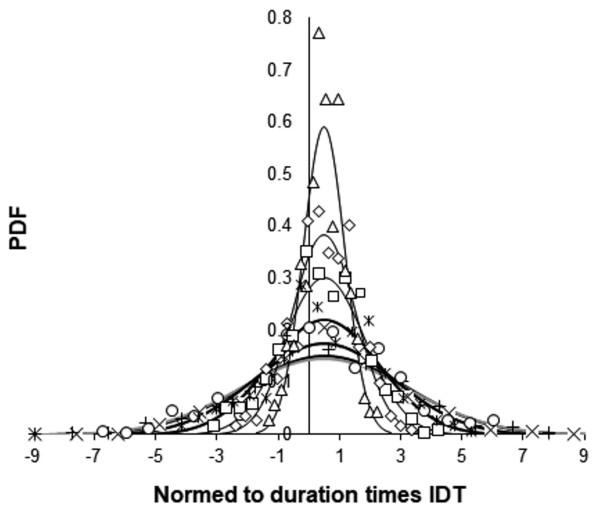

**Figure 6.** PDF of, normed to window length, IDT values calculated for consecutive 100 data windows of simulated noise data sequences, shifted by 100 data. From top to bottom black curves correspond to $\beta = 0.001$ (triangles), $\beta = 0.275$ (diamonds), $\beta = 0.545$ (squares), $\beta = 0.810$ (asterisks), $\beta = 1.120$ (circles), $\beta = 1.387$ (plus signs), and grey curve corresponds to $\beta = 1.655$ (cross signs).

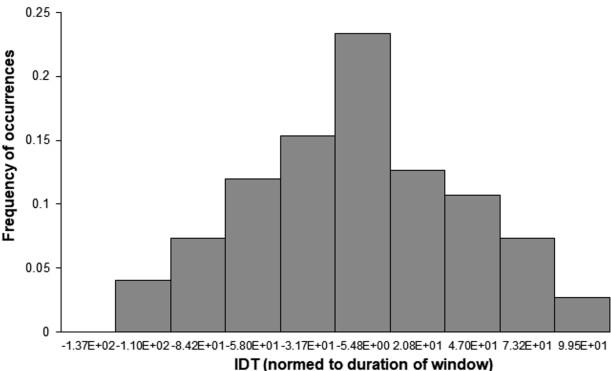

**Figure 7.** Frequency of occurrences of, normed to the span of window, integral deviation time values, calculated for each of 150 randomized catalogues for the whole period of duration.

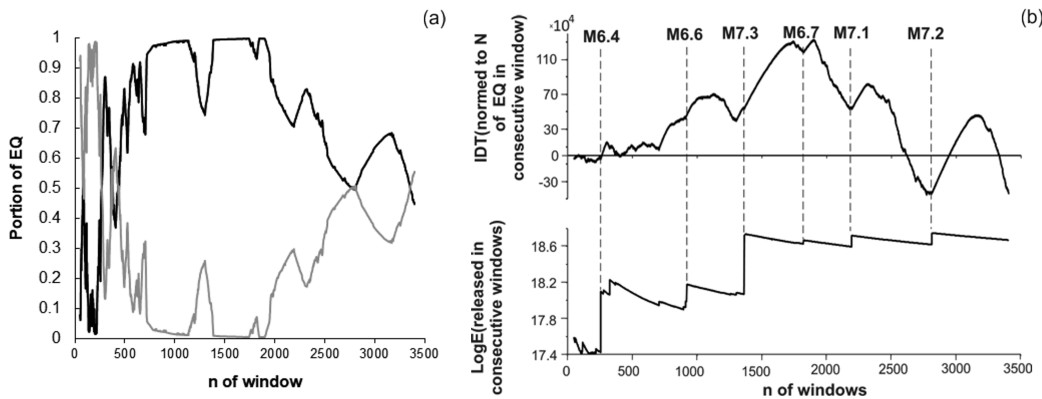

**Figure 8.** Calculated for extending by 10 consecutive data windows in the South California earthquake catalogue, a) portion of earthquakes occurred prior (grey) and after (black) regular markers in each window, b) normed to the number of EQs integral deviation times (top) and cumulative seismic energies (bottom)

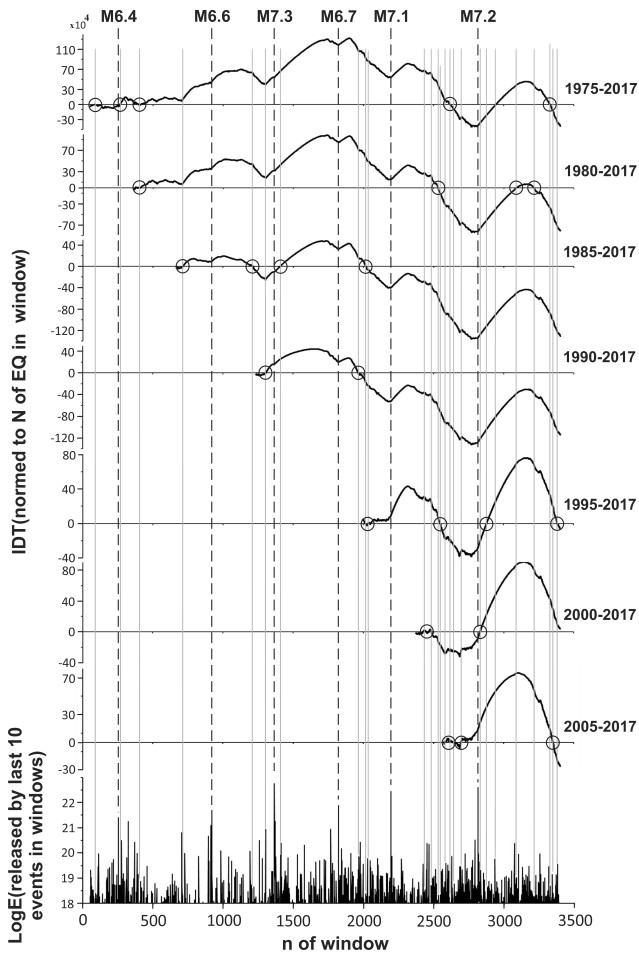

**Figure 9.** Calculated for the expanding (by consecutive 10 data) windows, integral deviation times (top 7 curves) and the increments of seismic energies released by 10 last events in consecutive windows (bottom curve) obtained from the south California earthquakes catalogue (above threshold M2.6). By the grey circles and grey vertical lines we show, where IDT curves cross abscissa axis. Dashed lines show the occurrence of largest earthquakes in the catalogue.

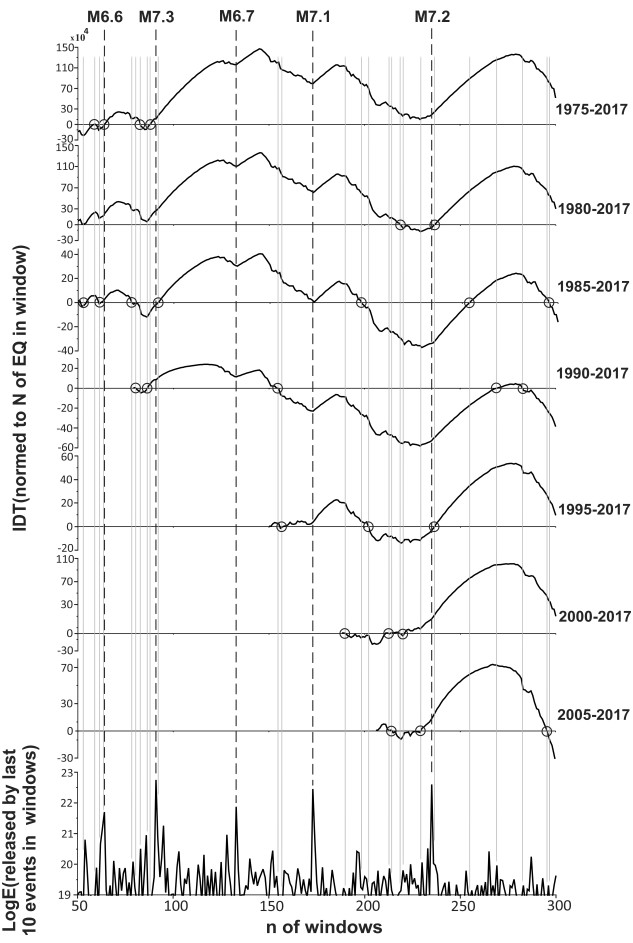

**Figure 10.** Calculated for the expanding (by consecutive 10 data) windows integral deviation times (top 7 curves) and increments of seismic energies released by 10 last events in consecutive windows (bottom curve) obtained from the South California earthquakes catalogue (above threshold M3.6). By the grey circles and grey vertical lines we show, where IDT curves cross abscissa axis. Dashed lines show the occurrence of largest earthquakes in the catalogue.

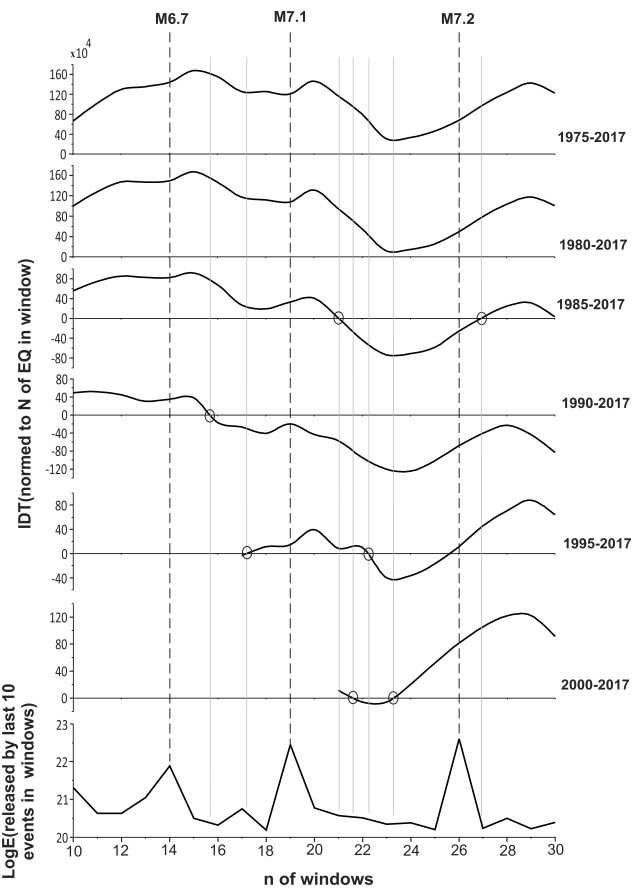

**Figure 11.** Calculated for the expanding (by consecutive 10 data) windows integral deviation times (here we have only top 6 curves, because for the $7^{th}$ curve, corresponding to the period 2005-2017 the number of events at the threshold M4.6 is small) and the increments of seismic energies released by 10 last events in consecutive windows (bottom curve) obtained from the south California earthquakes catalogue (above threshold M4.6). By the grey circles and grey vertical lines we show, where IDT curves cross abscissa axis. Dashed lines show the occurrence of largest earthquakes in the catalogue.

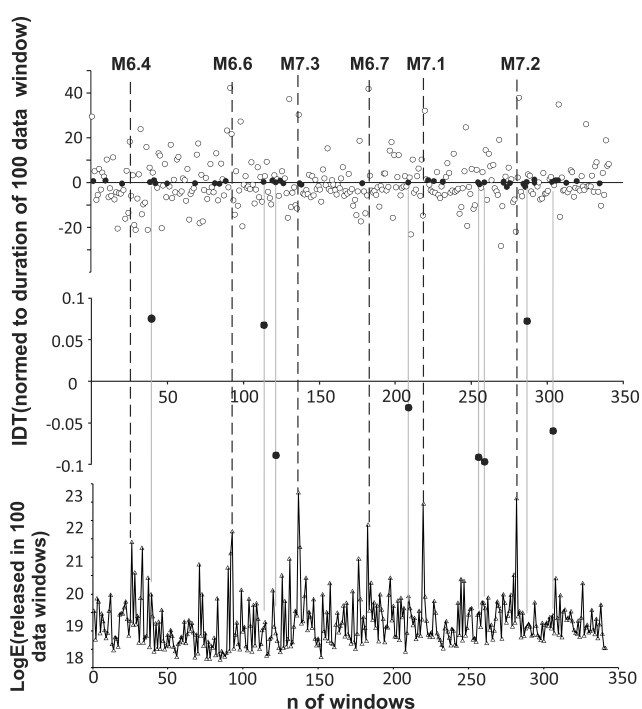

**Figure 12.** Calculated for the non-overlapping 100 data windows (shifted by 100 data), integral deviation times (circles in the upper and middle curves) and the released seismic energies (bottom curve). IDT values in vicinity of $0.1\sigma$ to zero are given by black circles in the top figure. IDT values in vicinity of $0.01\sigma$ to zero are given by black circles in the middle figure. By grey lines, we show location of closest to zero IDT values relative to the released seismic energy. Dashed lines show the occurrence of largest earthquakes in the south California catalogue (above threshold M2.6).

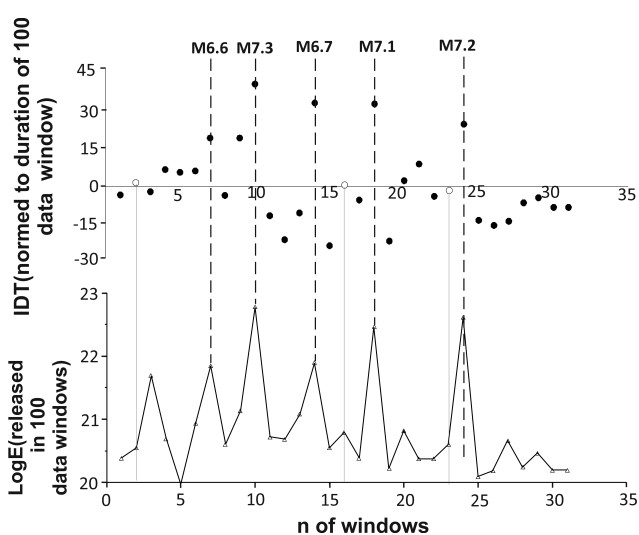

**Figure 13.** Calculated for the non-overlapping 100 data windows (shifted by 100 data), integral deviation times (circles in the upper curve) and the released seismic energies (triangle in bottom curve). IDT values in vicinity of $0.1\sigma$ to zero, are given by open circles in the top figure. By grey lines, we show location of closest to zero IDT values relative to the released seismic energy. Dashed lines show the occurrence of largest earthquakes in the South California catalogue (above threshold M3.6).