# Peer review of "Simple statistics for complex Earthquakes' time distribution"

_Nonlinear Processes in Geophysics, 2017_

## Referee Comment (RC1) · Anonymous Referee #1 · 23 Jan 2018

The authors study the southern Californian earthquake catalogue (1975-2017) analyzing the extend of regularity in the time series that is defined by the occurrence of earthquakes with magnitudes above 2.6. For that purpose they introduce the "integral deviation times" (IDT), a simple statistic measure that corresponds to the sum of the deviation times of the earthquake occurrences to regular times steps. As the authors state, the earthquake time distribution does not follow the patterns of a random process and there are several studies on the determination of the regularity of seismic processes and its changes in time. Yet, with regard to the presented IDT method I have several doubts concerning the appropriateness of that measure. Further, I see some weaknesses in the design of the analysis and clearness of the paper. At times it looks like you apply a bunch of methods without knowing why and what do you want to

show. The interpretation of the results could be more detailed and more related to the application, otherwise it is hard to see, what are the findings provided by the paper. In the following I comment on the mentioned shortcomings in more detail.

The motivation of the paper could be stronger. Why is it important to identify changes in the regularity of seismic activity? Do you expect to gain any knowledge for a better understanding of seismic processes? Do you expect to the give better predictions on earthquake occurrence based on changes in regularity? You should also refer to (some of) these questions in your conclusion. Also the provided background (domain) information could be more precise. Why do you consider only earthquakes with magnitude above 2.6 and after 1975? Please refer to the magnitude of completeness and possible changes in the time series due to improvements in recording. You should also report on the characteristics of seismic activity, e.g occurrence of cluster, foreshocks and aftershocks accompanying major earthquakes, assumption of iid (Poisson process) occurrence for declustered catalogues. It would be also nice to see a plot of (a part) of the time series, that e.g. illustrates the clustering of earthquakes in time. Also comment on why did you choose to study the southern Californian catalogue and clarify if there are any issues with induced seismicity.

As mentioned above, I have some doubts regarding the appropriateness of the IDT measure. On page 2 , line 16-17 you state IDT should approach zero for random sequences, if n goes to infinity. First, please correct the subsequent sentence, which says IDT approaches infinity for large n (I guess, this is a typo). Second, the statement needs to be proofed. Actually I doubt, that it is true. Let's assume the earthquakes would follow a Poisson process (purely random), the time series that is defined by the deviation times (DT), will be still highly autocorrelated. E.g. $P(DT(i)<0 \mid DT(i-1)<0) > P(DT(i)<0 \mid DT(i-1)>0)$ I calculated IDT for 100 Poisson processes with n=34020 events and an occurrence rate of 34020 / 22167178. The log value of absolute IDT/n was in 92 cases above 8, which is by magnitudes larger than the values calculated for colored noise in figure 5. In contrast, considering an equidistant time series (deterministic), DT

will be zero for each time step and consequently IDT will be 0.

In section 2 you explain several techniques for measuring regularity and show the results for applying those techniques to colored noise in section 3.1. This is a nice exercise, but I guess nothing new. What can you learn from those results and what do they tell you about the seismic time series in southern California? If you include these measures in your study, I would like to see them applied to the seismic time series. E.g. plot the power spectrum (figure 2) for the seismic data (which would be nice anyhow, to get a better impression about the real data) and plot the LZC, DET and CMSE values for the real data in figure 3 and 4. Regarding figure 5 you should also comment on the robustness of your results. Further, it would be helpful to provide some confidence intervals for IDT values of random processes. Actually, I am not sure, if you mix up things, since the IDT values I calculated for random processes are much higher. Do you calculate the sum/integral of deviation times from simulated noise data to regular time steps? Or do you calculate the sum/integral of the simulated noise data? Please, also check and comment on how comparable is the seismic time series to the simulated time series of colored noise.

In section 3.2 you generate randomized catalogues by shuffling the data, i.e. time and space locations and magnitudes (page. 7, line 20). I do not really understand, what you have done here. Since you do not consider space locations and magnitudes at that point of the paper, what is the effect of shuffling the data. The time steps do not change by shuffling, unless we have a different perception of the meaning of "shuffle". Please be more precise here. Apparently the time steps did change in your shuffled catalogues, otherwise you would receive the same IDT value for all catalogues. What can we learn/conclude from the consideration of the shuffled time series? It is not surprising that a randomized time series behaves more random, than a time series with interdependencies between the events.

What can we conclude from comparing the number of events prior (EQp) with those after (EQa) the regular time steps? Is the observed behavior typical for any kind of

time series (low/high frequency noise, tendency to cluster, . . .)? In figure 8 it looks like the fraction of EQp to EQa is quite random and could be completely different for similar seismic behavior (e.g. considering earthquakes from 1950 to 1975).

The results shown in figure 8 and 9 are not very surprising, Since earthquakes tend to cluster around main shocks (especially after large earthquakes, a large number of aftershock follows). Consequently at times of low seismicity the time steps between EQs are larger and the EQs will tend to occur after the regular time steps, which leads to negative DT values (if DT(i) = T_R(i) − T_EQ(i)) and decreasing IDT. At times of high seismicity (especially after large earthquake) the time steps between the EQs become shorter and EQs will tend to occur prior to the regular time steps, which leads to positive DT values and increasing IDT. I would need a more in depth analysis and interpretation of the results, to get any new information. For example, I would like to see the calculation of the other regularity measures introduced section 2 on the real data set and a comparision with IDT values. Also you should consider to apply your method on earthquake catalogues of different regions. Considering the results presented in that paper, I have no idea what to expect. I might get a better understanding of the presented IDT approach, if results from other catalogues are compared to the southern California results. You might also study the behavior of IDT in periods of induced seismicity (e.g. Oklahoma).

Some statements would need a statistic test/proof to be more than a subjective judgement. E.g. page 9, line 6-7: "lower IDT value corresponds to period with decreased sesimic activity". In figure 9, the IDT values around M6.4 and M7.2 as well as in figure 10 the IDT values around M6.6, M7.3 and M7.2 are quite small compared to the other IDT values. In fact, large earthquakes are rather close to local minima of IDT values. Page 10, line 11-13: "close to zero values of IDT can be regarded as random". This needs to be proofed. "[. . .] they occur in periods of decresed seismic energy release" This seems to be subjective perception. It is hard to see, but e.g. the energy release for the first and third point is not that small. I agree, that the very small IDT values do

not coincide with the large earthquakes, but the chance of coincidence is also quite small.

It is a good idea to compare the behavior of time series with different threshold magnitudes. To include more observations for larger magnitudes, you should consider to increase the considered time span. Since larger earthquakes are easier to detect, time series that start before 1975 can be considered (again, refer to magnitude of completeness).

Minor issues:

You should define DT(i). DT(i) = T_EQ(i) – T_R(i) or DT(i) = T_R(i) - T_EQ(i)

Please use scientific format (x*10^n) for your numbers. It is quite cumbersome to count the number of digits to be able to compare the provided numbers.

Figure 6: It would be more intuitive to plot a histogram for frequencies, instead of a continuous function. Otherwise explain the meaning of the dots and how you derive the function.

Figure 7: Please use a Y-axis starting with 0. Also, please use intuitive x labels (e.g. SDTa and SDTp).

You should comment on how you determine the energy release and what is the energy release (relation to magnitude).

Figure 9: Why do you highlight the points where the IDT curve crosses the abscissa axis? What is the meaning of these points?

When considering shortened time series (e.g. figure 9 – 11), you should take care to also adapt the regular time series to the length and rate of the corresponding time series (otherwise you change your definition of IDT).

Page 14, line 7-9: It is very natural that a fraction of points is within one tens of the standard deviation.
Language should be improved. Especially, sentences starting with "Exactly" should be replaced with something like "To be (more) specific/precise", "In detail", ...

---

## Referee Comment (RC2) · Anonymous Referee #2 · 31 Jan 2018

GENERAL COMMENTS The authors describes a simple statistical methods to evaluate the time series distribution of earthquakes picked up from the Californian Earthquake Data Center.

SPECIFIC COMMENTS They limit the study since 1975, why? The Catalog reports data since at least 1932. They select the earthquake's magnitudes greater than 2.6, moreover they do not make distinctions between depths of hypocenter. The authors don't even identify the spatial region, they simply took the data in the archive taken without criticism. They don't select the main shock from aftershocks. So the statistical description and the results are affected by these undefined choices. The Conclusions are trivial.

CONCLUDING REMARK The goals of the work are not well motivated; it seems to be

a mere statistical exercise. So the paper needs a deep afterthought.

---

## Author Comment (AC1) · 14 Mar 2018

Dear reviewer, let us express our sincere gratitude for your work and competent evaluation of our manuscript. In our opinion most of your questions are answered below or we explain our vision of certain questions. Also, let us inform you that as far as the same or almost similar questions are repeated along in the reviewers text we apologize that could not avoid some repetitions in our answers too.

*Anonymous Referee #1*

*The authors study the southern Californian earthquake catalogue (1975-2017) analyzing the extend of regularity in the time series that is defined by the occurrence of earthquakes with magnitudes above 2.6. For that purpose they introduce the "integral deviation times" (IDT), a simple statistic measure that corresponds to the sum of the deviation times of the earthquake occurrences to regular times steps.*
*As the authors state, the earthquake time distribution does not follow the patterns of a random process and there are several studies on the determination of the regularity of seismic processes and its changes in time. Yet, with regard to the presented IDT method I have several doubts concerning the appropriateness of that measure. Further, I see some weaknesses in the design of the analysis and clearness of the paper.*

*At times it looks like you apply a bunch of methods without knowing why and what do you want to show.*

We thank reviewer 1, for this remark. We would like to underline that we definitely know why several ("bunch" of) contemporary methods of data analysis have been used in the present work. These well known and often used methods (LZC, RQA, CMSE) are very effective tools, when correctly used, for the task of quantification of dynamics of complex processes: examples can be easily found in a number of articles from different fields. We regret that, as it appears, what we wanted to show using certain data analysis methods was not clear in the previous version of the manuscript. In the corrected version, it is underlined that these methods have been used for the very important for our research task. Namely, it was required to ensure that the simulated random data sequences (here different type of noises and Poisson process data sets), being generally complex and random-like, are still different in the sense of underlying dynamics and that these differences are quantifiable. The problem is that the simulated noise datasets, by the conditions of their generation, should differ by the features of their frequency content. At the same time, it was necessary to know if these data sets are different in the sense of regularity, especially at small differences between spectral exponents. Here, it was necessary to assess the extent of regularity in

noise data sets from different point of views, i.e. use analysis methods based on different underlying principles. For our research purposes such testing by standard analysis tools was absolutely necessary step prior to proceed to the analysis of the same simulated data sets by IDT. Next, we needed to be convinced that such a simple statistical method like IDT can discern differences in dynamical features of complex high-dimensional processes (differences in which already have been documented by standard complex data analysis methods). This is why we spent considerable part of our time and carefully compared results of IDT analysis with the results of other, well known and many times critically tested, methods (here LZC, RQA and CMSE). These analyses bring us to the conclusion, that results of IDT are in principal agreement with the results of used standard tools of complex data analysis.

Here, for readers who are not so aware of the details of modern complex data analysis, the following question may arise - if the results of IDT agree with those obtained by some other methods, why do we need to develop a new tool giving similar conclusions? Also, it indeed may be a need of an additional explanation why standard methods have not been used together with IDT calculation for the real (obtained from earthquake catalogue) data sets. At first, we should state that each of methods are developed to test data sets from a certain point of view, e.g. LZC is based on information theory, RQA -on phase space population testing, CMSE – on entropy assessment, etc. Moreover, all the complex data analysis methods (used here and others) to be correctly used necessitate special conditions to be fulfilled both in the sense of quality and length of data sets, as well as in the sense of calculation purposes (e.g. conditions for reliable phase space reconstruction or coarse-grained series construction). Therefore, knowing weak and strong sides of these methods for certain data sets, we additionally wanted to have a testing method based on the very simple statistical and distributional features of complex process (data sets). This was interesting to get in this way the possibility to look at the complex process from a simple new point of view, which will not be complicated by the fundamental principles of method's accomplishment. Such simple vision definitely has its own restrictions and, as in the case of any other data analysis methods, should be used correctly. Anyway, as it follows from our results, proposed calculation method is effective for data sets (especially for short ones), simulated from complex processes as well as original and randomized data sets of earthquake's time distribution. Such test is very important for the usually not perfect quality data sets of the real measurements. It should be also pointed out that the IDT calculation method has no practical restrictions on the length of data sets because of its simplicity. (we mean statistically reasonable length of data sets of at least several tenth of data).

*The interpretation of the results could be more detailed and more related to the application, otherwise it is hard to see, what are the findings provided by the paper.*

Together with the presentation of a simple and effective method for complex data sequences analysis, in this manuscript we present the results of its application for the time distribution of earthquakes taken from the south Californian catalogue. Main finding of this work is a clear quantitative demonstration that the extent of regularity of earthquakes time distribution is changing over the time. It was shown that, over the period of analysis, we can indicate periods when earthquakes' time distribution became most random as well as those when it is less random. Such a finding for the seismic process, in our opinion, is of immense importance, as far as by many

authors seismic process still is regarded as completely random, i.e. not having a quantifiable dynamical structure (unpredictable). Most important is that the extent of randomness never reaches its maximum in periods immediately prior to strongest earthquakes. This points to the increase of determinism in earthquake generation process (at least in temporal domain) and thus makes researches aimed at finding of the precursory markers for strong earthquakes in the complex seismic process, a well-grounded scientific task.

*In the following I comment on the mentioned shortcomings in more detail. The motivation of the paper could be stronger. Why is it important to identify changes in the regularity of seismic activity? Do you expect to gain any knowledge for a better understanding of seismic processes? Do you expect to the give better predictions on earthquake occurrence based on changes in regularity? You should also refer to (some of) these questions in your conclusion.*

The main general problem targeted in our last researches concerns with the dynamics of seismic process. Exactly, investigation of features of earthquakes time distribution is often posed important task not only for us, but also for many research groups worldwide for last decades [e.g. Davidsen, C. Goltz, 2004; Kawamura. 2007; Kenner, M. Simons, 2005; etc.]. Motivation for the present work was to assess how the extent of regularity in the earthquakes time distribution changes over the considered period of catalogue time span. It needs to be underlined that, in spite of the above-mentioned and many other studies, the problem of how regularity of seismic process is changed still remains unanswered. At the same time, it is clear that without such knowledge the better understanding of seismic processes can not be achieved. Moreover, scientific posing of such general tasks as earthquake prediction or control of seismic processes, will not be look grounded unless basics (at least main) of features of its dynamics in spatial temporal or energetic domains will not be understood.

*Also the provided background (domain)
information could be more precise. Why do you consider only earthquakes with magnitude
above 2.6 and after 1975? Please refer to the magnitude of completeness and
possible changes in the time series due to improvements in recording. You should
also report on the characteristics of seismic activity, e.g occurrence of cluster, foreshocks
and aftershocks accompanying major earthquakes, assumption of ii d (Poisson
process) occurrence for declustered catalogues.*

As it is said in the revised version of manuscript, we aimed to analyze temporal features of the original (natural) process of earthquake's generation. For this purpose, we selected a best quality catalogue of southern Californian seismic activity (Fig.1). Being aware of the problems that can be caused by the inappropriate "bleaching" of complex data sets [e.g. Abarbanel, 1993], and aiming at the analysis of temporal features of seismic process, we would like, when it is possible, to avoid any cleaning, filtering or declustering of catalogue in order to preserve its original time structure. Consequently, we tried to have as possible long period of observation with as possible low representative threshold. For this purpose, according to results of time completeness analysis

(Fig. 3) we decided to be focused on the time period from 1975 onward. Indeed, we see that since the middle of 70th of last century Mc was clearly decreased what finally enabled us to work with the southern Californian earthquake catalogue at the representative threshold M >= 2.6, according to the Gutenberg–Richter relationship analysis (see Fig. 2). We understand that in such catalogue we deal with both, independent as well as dependent (aftershocks or foreshocks) events. Presence of both type of events in the catalogue looked for us quite acceptable in the frame of research task because here we speak about the general features of time behavior of seismic process and also because the physics of generation of dependent and independent events is similar [Davidsen and Goltz, 2004; Martinez, et al. 2005]. In any case to assess possible influence of dependent events on the results of our calculations, we performed analysis at higher representative thresholds M3.6, M4.6 and even for M5.6, when this was possible because of small number of events. According to our analysis dependent events do not essentially influence results of IDT analysis.

[Figure]

Fig. 1.

[Figure]

Fig. 2.

[Figure]

Fig. 3.

*It would be also nice to see a plot of (a part) of the time series, that e.g. illustrates the clustering of earthquakes in time.*

We analyzed the original (not filtered) catalogue (Fig.1) with obvious natural clustering in different domains. The time clustering in this catalogue is well known and described for many times (also in one of our previous article Matcharashvili et al, in Physica A 433 (2015). Thus, the earthquakes time clustering in Californian catalogue is obvious for our research period and in our opinion there is no need for additional illustrations. This is why we do not show interevent time series, which was possibly meant by the reviewer 1.

*Also comment on why did you choose to study the southern Californian catalogue and clarify if there are any issues with induced seismicity.*

As mentioned above, we have used southern Californian catalogue for its high quality. Having such best quality catalogue, we do not aimed to go further in the analyses of effects like of induced seismicity. At the same time, testing carried out at increased representative thresholds apparently shows that their influence on IDT calculation results can be regarded as negligible.

*As mentioned above, I have some doubts regarding the appropriateness of the IDT measure. On page 2 , line 16-17 you state IDT should approach zero for random sequences, if n goes to infinity. First, please correct the subsequent sentence, which says IDT approaches infinity for large n (I guess, this is a typo). Second, the statement needs to be proofed. Actually I doubt, that it is true.*

Sorry, but it is not clear where the typo is. In our text it is said: "the sum of the deviation times should approach zero in the infinite length limit". As we understand in common parlance, this means that IDT approaches zero if number of deviations is large enough.

Generally the question of close to zero IDT values was partly discussed above and here we add some following thoughts. Logically IDT should approach zero for random sequences, if $n$ goes to infinity, and empirically for sequences closer to randomness we indeed get IDT closer to zero comparing to less random sequences. Presented in the revised version of manuscript new analysis and results obtained after reviewer's remark confirm this statement. Theoretical basics of IDT will be given in the next article in collaboration with our colleague Prof. Czechowski from Institute of Geophysics, Warsaw, Poland. Thus, in the present work we decided to be restricted by strong empirical argumentation on the certain data sets. As an example of sequences with different extent of randomness we used the series of color noise data sets. In general, there may be many different random sequences and the question about which out of these random sequence is "more random" and which is "less random" is not easy to answer. Therefore, according to our purpose (explained above) and to have strong arguments, why we regard some sequences as more and others as less random, in this research we used well known and accepted methods of complex data analysis like PSR, LZC, RQA and CMSE.

Thus, as it is mentioned in the manuscript, we generated artificial noise data sets, which, as it was shown, are quantifiably different – i.e. represent different types of colored noises (in the revised version we consider 7 simulated data sets instead of 8 in the former version, as far as noise data set with $\beta = 1.932$ gave result very similar with $\beta = 1.655$). Here it is necessary to emphasize that in order to make the simulated data sets closer to a character of the temporal evolution of seismic process, we used the sequences of positive numbers.

As we explained in the manuscript, as well as here above, we needed to have data sets with reasonable differences in the extent of regularity in order to find out, whether calculation of IDT may be sensitive to the dynamic changes taking place in the analyzed data sets. We agree with reviewer that, as far as we aimed to use IDT for seismic data sets, for method testing purpose, it was indeed more logical to consider also random process, which is often used by seismologists – Poisson process. In the present version of manuscript, we added results for Poisson processes with different lambda values, obtained by used methods. Results of analysis is described in the revised manuscript. From these results, we see that the conclusion drawn from a simulated noise data analysis, that more random process gives closer to zero IDT value, is correct for Poisson processes

too. In other words, Poisson process and white noise look similar according to results of PSR, LZC, RQA and CMSE analysis. It is interesting that, by IDT results we see that Poisson process looks even more random than simulated data set, which is closest to a white noise.

Only explanation of why reviewer 1 got IDT for Poisson process "by magnitudes larger than the values calculated for colored noise" is apparently connected with the procedure of norming. In order to avoid possible misunderstandings caused by different "time span of window", IDT should be normed to "time span" (cumulative sum) values or compared to calculation accomplished for the same time span - in case of reviewers example, to IDT from the original catalogue.

For further clarity, regarding IDT values for different noises, in the revised version we add pdf curves in Fig.6. For Poisson process, IDT results is not shown in this figure because it is close to white noise.

*Let's assume the earthquakes would follow a Poisson process (purely random), the time series that is defined by the deviation times (DT), will be still highly autocorrelated. E.g. P(DT(i)<0 | DT(i-1)<0) > P(DT(i)<0 | DT(i-1)>0) I calculated IDT for 100 Poisson processes with n=34020 events and an occurrence rate of 34020 / 22167178. The log value of absolute IDT/n was in 92 cases above 8, which is by magnitudes larger than the values calculated for colored noise in figure 5.*

Once you generated Poisson data sets by the condition N=34020 span=22167178, it was more correct to compare the result with real seismic data sets, e.g. Fig.8 (IDT=-14611458375), from which we see that IDT of data set from the catalogue is about three orders of magnitude larger. We emphasize that in this case N and span of these data sets is similar and corrections for differences in the time span is not necessary. On the other hand, when we compare IDT of your Poisson sequences, with the results for color noises, it is necessary to make corrections because of the differences in the time span, i.e. you need to norm the IDT values to the "time span or range", which is different for color noises and Poisson data sets. After norming you will see that from IDT point of view there is no difference between color noises and Poisson sequences (or Poisson process gives somehow smaller IDT values than color noises) and this is logical because we deal generally with random sequences, which may be just slightly different.

*In contrast, considering an equidistant time series (deterministic), DT will be zero for each time step and consequently IDT will be 0.*

As it was already explained we do not consider the case, when equidistantly distributed over given time interval data set is compared with the sequence of regularly distributed over the same time period markers, this has no sense. This is a prerequisite of the presented method that when it is possible (in the physical world it is practically always), the original sequence and sequence of time markers should follow different features of time evolution. Otherwise we got simply IDT=0 (at least statistically for the set of different time markers with the same distribution features).

In this respect, we repeat the general idea of IDT here. We aimed to analyze the character of EQ time distribution and compare it with the sequences of markers that are distributed over the

same time interval according to the predefined distributional features. We are working to develop an analysis tool based on this idea for different time marker sequences (with different distributional features) in our ongoing research. In the present work, in the frame of aforementioned general view, it was logical to start from the comparison of EQ catalog data with the sequence in which time markers are distributed regularly.

*In section 2 you explain several techniques for measuring regularity and show the results for applying those techniques to colored noise in section 3.1. This is a nice exercise, but I guess nothing new.*

We think that analysis of simulated data sets should not be regarded as a mere exercise, but viewed in the context of targeted research. Indeed, as we mentioned above, we needed to fulfill analysis on simulated complex data sets with predefined different extent of randomness. Only after such analysis and appropriate data selection, we could undoubtedly prove that IDT is able to discern and quantify the changes even in the case, when we deal with short data sets from a complex process. So, this analysis was a necessary part of research aimed to present and launch the new method of IDT. Besides, in our opinion, the results obtained from the careful analysis of different simulated random data sets, given compactly in one article, will be undoubtedly helpful for researchers from different fields for different testing purposes.

*What can you learn from those results and what do they tell you about the seismic time series in southern California?*

We mentioned above that analysis on simulated data sets was a necessary step to conclude that IDT is sensitive to dynamical changes in complex data sets and especially even in the case of short data sets. About conclusions on seismic process drawn from the IDT analysis we already described above.

*If you include these measures in your study, I would like to see them applied to the seismic time series. E.g. plot the power spectrum (figure 2) for the seismic data (which would be nice anyhow, to get a better impression about the real data) and plot the LZC, DET and CMSE values for the real data in figure 3 and 4. Regarding figure 5 you should also comment on the robustness of your results.*

In this work our interests are focused on dynamical changes occurred in small data sets (also obtained from seismic catalogue) and on the development of appropriate for this task analysis method. The reason why we needed to use IDT approach is given above where we explained that LZC, DET and CMSE are not developed for very short data sets used in our research (windows of 100 data span). At the same time, we base our main conclusions on the results obtained for short data sets. Also, we should state again that we work with a specific process of time evolution of earthquake occurrences. In this case we deal with strong trend which usually complicate using of standard data analysis tools. Thus, we do not show results of LZC, DET and CMSE calculation for short and very short sequences.

On the other hand, in order to somehow fulfill the reviewer's interest to the use of complex data analysis tools to seismic process, we present here calculations for the entire length data sets of interevent time sequences: LZC =0.71, %DET=35.

[Figure]

Power spectrum of original (top) and shuffled (bottom) waiting times intervals sequence from the Southern Californian catalogue 1932-2013. (from Matcharashvili et al. Physica A, 2015).

[Figure]

CMSE values versus scale factor for interevent data sequences.

All these results obtained for a whole catalog show just a trivial fact that we deal with a complex seismic process. For reliable quantification of dynamical changes in such processes, especially occurring on the small time scales, we need to use specially developed methods, e.g. like used here IDT test.

*Further, it would be helpful to provide some confidence intervals for IDT values of random processes. Actually, I am not sure, if you mix up things, since the IDT values I calculated for random processes are much higher. Do you calculate the sum/integral of deviation times from simulated noise data to regular time steps? Or do you calculate the sum/integral of the simulated noise data? Please, also check and comment on how comparable is the seismic time series to the simulated time series of colored noise.*

We thank reviewer for this comment. In the revised version we calculated IDT values for sliding windows of 1000 and 100 data and calculated averaged values. These sequences of IDT values

calculated for 1000 and 100 data windows of simulated data sets then have been compared by paired sample *t* test and significant differences at p=0.01 have been demonstrated. This is mentioned in the revised version.

Noise data sets consisted of positive values and thus generally did not contradict to the physical meaning of the time evolution in original data sets. Additionally, according to the reviewer's suggestion, in the revised version we added also Poisson process data sets as far as this process is often used in the context of seismicity.

*In section 3.2 you generate randomized catalogues by shuffling the data, i.e. time and space locations and magnitudes (page. 7, line 20). I do not really understand, what you have done here. Since you do not consider space locations and magnitudes at that point of the paper, what is the effect of shuffling the data. The time steps do not change by shuffling, unless we have a different perception of the meaning of "shuffle". Please be more precise here. Apparently the time steps did change in your shuffled catalogues, otherwise you would receive the same IDT value for all catalogues. What can we learn/conclude from the consideration of the shuffled time series? It is not surprising that a randomized time series behaves more random, than a time series with interdependencies between the events.*

As it is said in the revised version, in order to see whether obtained from the original catalogue IDT value is the characteristic of time distribution of natural seismic process or is caused by unknown random effects we started to calculate IDT values for the set of randomized catalogues. Such comparisons are often used in the context of surrogate data testing in complex data analysis. In these artificial catalogues the original time structure of the southern Californian earthquakes distribution was preliminary destroyed. More precisely, the occurrence time of the original events has been randomly shuffled (i.e. earthquakes' time locations have been randomly changed over the more than 42 year of considered period). We have generated 150 of such randomized catalogues and for each of them, IDT values have been calculated for the whole catalogue time span (what was the same as for the original catalogue). Thus, to generate randomized catalogues, we used other method distinct from just shuffling of interevent times. We regret that in the former version of manuscript by mistake it was said that randomization was accomplished in the time, space and energetic domains, which we plan to do in the next works. Randomization based on randomly rearranged occurrence times of earthquakes in the original catalogue was described also in Matcharashvili et al. Physica A 2015.

*What can we conclude from comparing the number of events prior (EQp) with those after (EQa) the regular time steps? Is the observed behavior typical for any kind of time series (low/high frequency noise, tendency to cluster, : : :)?*

In the present version we consider the differences in the number of earthquakes occurred for the entire observation time, prior and after of the corresponding regular markers. We also decided to carry out additional calculation of summary deviation times separately for each of these groups. The sum of deviation times, normed to the number of corresponding earthquakes prior or after regular markers are essentially smaller in the case of randomized catalogues than for original

catalogue. Though this again confirms that in the case of random sequence IDT is closer to zero but finally we agreed with reviewer and decided that in revised version there is no need in such additional arguments.

*In figure 8 it looks like*
*the fraction of EQp to EQa is quite random and could be completely different for similar*
*seismic behavior (e.g. considering earthquakes from 1950 to 1975).*

In Fig. 8a of the revised version, we show just an example of the variation of portion of earthquakes occurred prior (grey) and after (black) regular markers for the observation windows. It is clear that this picture will be changed for other time periods (catalogue time span) or areas of location.

*The results shown in figure 8 and 9 are not very surprising, Since earthquakes tend*
*to cluster around main shocks (especially after large earthquakes, a large number of*
*aftershock follows). Consequently at times of low seismicity the time steps between*
*EQs are larger and the EQs will tend to occur after the regular time steps, which leads*
*to negative DT values (if DT(i) = T_R(i) – T_EQ(i)) and decreasing IDT.*

In our opinion, IDT will not always decrease when the number of earthquakes decrease at relatively seismically quite periods. This will depend on the distribution of events relative to regular markers. More expectable seems that when the number of events (which are functionally connected with independent main shocks) decreases, the probability that these events will be more or less symmetrically distributed on both sides of regular markers will be larger, than in the case of functionally strongly connected (correlated) events. In the last case, asymmetric (relative to regular markers) distribution seems to be more probable. This looks quite logical at least because time intervals between less interconnected earthquakes should be statistically larger than that for correlated events.

*At times of*
*high seismicity (especially after large earthquake) the time steps between the EQs*
*become shorter and EQs will tend to occur prior to the regular time steps, which leads*
*to positive DT values and increasing IDT.*

    I guess here negative DT and decreasing IDT is meant.
    We underline that, here and above, we do not question a trivial fact that time steps between EQs at lower seismicity rate may be longer and that time steps between aftershocks of large earthquakes would be apparently shorter. What we show is that time distribution of earthquakes at lower seismicity is more random than for aftershock activity after strong events. This is quite logical that time evolution of functionally dependent from the main shock aftershocks will be more deterministic- regular, than those not strongly connected with other events. Main result of our work is that now we clearly show that this logical conclusion can be proven quantitatively.

 *I would need a more in depth analysis and*
*interpretation of the results, to get any new information. For example, I would like to see*

*the calculation of the other regularity measures introduced section 2 on the real data set and a comparision with IDT values.*

New in this work are two things. First - a demonstration that such a simple statistics like presented here may be useful for complex processes analysis like time evolution of seismic process. Second - the quantitative documentation that the regularity of the time distribution of earthquakes is changing over time and that it is more regular at lower seismic activity than in periods of strong earthquakes occurrences. To our knowledge this is indeed new information. We'd appreciate if the reviewer can suggest references with direct indication of such kind.

Presented analysis is so simple that it can be critically tested by anyone with basic knowledge in statistics (if analysis will be done correctly). As for using standard methods for time distribution of earthquakes we state again that there are two reasons why we did not show results of such analysis. First is the quality of seismic data sets and inappropriateness of these methods for very short sequences. Second is specificity of used time evolution of earthquakes as data sets. It is really not easy to imagine how for such data, with strong trends, methods of complex data analysis can be used unless these data preliminary will be somehow handled (noise reduction, filtering, etc.). But all such procedures will destroy original dynamics of seismic process what we would like to avoid. On the other hand, if we go to the interevent sequences as logical alternative of data sets in the context of time distribution analysis, then we should realize that this is not the same as real time evolution of process. Knowing that such problems may arise we, in the first part of manuscript, tried to analyze effectiveness of method for the set of carefully simulated and tested data sets with known and quantified changes in dynamical structure.

*Also you should consider to apply your method on earthquake catalogues of different regions. Considering the results presented in that paper, I have no idea what to expect. I might get a better understanding of the presented IDT approach, if results from other catalogues are compared to the southern California results. You might also study the behavior of IDT in periods of induced seismicity (e.g. Oklahoma).*

We definitely have such plans to do analysis on different catalogues in the frame of our future works but do not want to make present article too large. Here we mostly care to present method on the example of trustworthy catalogue and discuss new results on earthquakes time distribution in California.

*Some statements would need a statistic test/proof to be more than a subjective judgement.*
*E.g. page 9, line 6-7: "lower IDT value corresponds to period with decreased sesimic activity".*

Apparently, the reviewer means the sentence on page 13, when we comment results presented in Fig. 10 and 11. We again state that in these figures grey vertical lines cross Log E curve exactly at points where in most cases seismic energy release decreases–by about two orders comparing to observed maximums of energy release. The only way to do statistical analysis for this kind of data sets is to compare them with the time evolution and energy release in the randomized catalogues.

In the present version we mention about significant difference between original and time randomized cases.

*In figure 9, the IDT values around M6.4 and M7.2 as well as in figure*
*10 the IDT values around M6.6, M7.3 and M7.2 are quite small compared to the other*
*IDT values.*

Indeed, according to our results at decreased energy release, IDT values are smaller.

*In fact, large earthquakes are rather close to local minima of IDT values.*
*Page 10, line 11-13: "close to zero values of IDT can be regarded as random". This*
*needs to be proofed.*

In order to prove the fact that "close to zero values of IDT can be regarded as random" we present results in the section 3.1 "Analysis of model data sets" as well as in next section. All this  in our opinion is convincing.

*"[: : :] they occur in periods of decresed seismic energy release"*
*This seems to be subjective perception. It is hard to see, but e.g. the energy release*
*for the first and third point is not that small.*

We are grateful for this remark. We are sorry for mistake in legends of figures 9-11 in the former version of manuscript. Indeed, it is much more correct to say that the amount of the seismic energy released by the last 10 events of expanding windows decreased; this is shown in the lower curve of Fig.9.
So, the situation in Figs. 9, 10 and 11, indeed looks confusing as far as from one side we present IDT values calculated for expanding windows and on the other side we show seismic energy which is calculated for just the last 10 earthquakes in each expanding windows. It can be assumed that the solution for better visibility here is to come back to the form of energy release presentation like in Fig.8 (where IDT and energy are calculated for the same windows), but in this case we do not see fine structure of changes in energy release against the background of the summary amount (over the whole window) of the energy release. This is why we finally decided that the present form of Figs. 9.10 and 11 is more informative in the sense of better visibility of location of windows, in which the amount of released seismic energy  tends to decrease and the process of earthquakes distribution become more random (in the sense already shown for simulated data sets and randomized catalogues) - the curve of IDT values cross the $x$ axis. To make situation more convincing and in order to test results obtained for expanded windows, we accomplished additional analysis for the fixed length data windows. In Figs 12 and 13, energy and IDT values are calculated in the same size (100 data) windows. Results obtained for both fixed size and expanded windows shows that decrease in the local amount of seismic energy occurs in windows where IDTs are closer to zero, comparing to other windows.

I agree, that the very small IDT values do not coincide with the large earthquakes, but the chance of coincidence is also quite
small.
It is a good idea to compare the behavior of time series with different threshold magnitudes.
To include more observations for larger magnitudes, you should consider to
increase the considered time span. Since larger earthquakes are easier to detect, time series that start before 1975 can be considered (again, refer to magnitude of completeness).

We give the above explanation why a certain catalogue from 1975 to 2017 is used at M2.6 representative threshold. This catalogue is in complete accordance with our goals in the present research. Anyway, in order to express our gratitude to the reviewers hard work to improve present manuscript, we present here results of our analysis for the south Californian catalogue from 1932 to 2017 at representative threshold M3.5. It is clearly visible that the main conclusions from used catalogue (M2.6) are confirmed for longer time period and higher threshold values. IDT values in the range from $0.1\sigma$ to zero were found in the periods with relatively low (two- three orders lower than observed maximums) seismic energy release. At the same time, in the present work we do not intend to highlight these similar results, which definitely would be discussed in our next work in the near future.

[Figure]

Fig. 5.
Calculated for the non-overlapping 100 data windows (shifted by 100 data), integral deviation times and the released seismic energies (bottom curve). IDT values in vicinity of $0.1\sigma$ to zero, are given by red squares. South Californian catalogue 1932-2017, M3.5.

Minor issues:
You should define DT(i). DT(i) = T_EQ(i) – T_R(i) or DT(i) = T_R(i) - T_EQ(i)
Please use scientific format (x*10ˆn) for your numbers. It is quite cumbersome to count the number of digits to be able to compare the provided numbers.

Figure 6: It would be more intuitive to plot a histogram for frequencies, instead of a continuous function. Otherwise explain the meaning of the dots and how you derive the function.

We changed Fig. 6 according to the reviewer's suggestion. Also, we omitted former Fig.7, and added PDF of normed to the window duration time IDT values calculated for consecutive 100 data windows of simulated noise data sequences, shifted by 100, which in our opinion is more informative.

Figure 7: Please use a Y-axis starting with 0. Also, please use intuitive x labels (e.g. SDTa and SDTp).
  Done.

You should comment on how you determine the energy release and what is the energy release (relation to magnitude).

We give now reference according to which seismic energy was calculated from magnitudes [Kanamori,1977].

Figure 9: Why do you highlight the points where the IDT curve crosses the abscissa axis? What is the meaning of these points?

We just wanted to make better visible for readers location of crossing points (i.e. situation when IDT comes closer to zero).

When considering shortened time series (e.g. figure 9 – 11), you should take care to also adapt the regular time series to the length and rate of the corresponding time series (otherwise you change your definition of IDT).

We agree with the reviewer about importance of norming. Anyway, in this case, we are mostly interested in the question, whether and when IDT comes closer to zero, what (in this case) is not influenced by the norming procedure.

Page 14, line 7-9: It is very natural that a fraction of points is within one tens of the standard deviation.

Most importantly, these points (IDT values) correspond to windows with lower release of seismic energy.

Language should be improved. Especially, sentences starting with "Exactly" should be replaced with something like "To be (more) specific/precise", "In detail", ...

---

## Author Comment (AC2) · 14 Mar 2018

GENERAL COMMENTS The authors describes a simple statistical methods to evaluate the time series distribution of earthquakes picked up from the Californian Earthquake Data Center.

SPECIFIC COMMENTS They limit the study since 1975, why? The Catalog reports data since at least 1932. They select the earthquake's magnitudes greater than 2.6, moreover they do not make distinctions between depths of hypocenter.

As it is underlined in the revised version of manuscript, we aimed to analyze temporal features of original earthquakes generation process. For this, we selected the best quality catalogue of southern Californian seismic activity (Fig. 1). Knowing problems, which can be caused by inappropriate "bleaching" of complex data sets [e.g. Abarbanel, 1993], in this work aiming at the analysis of temporal features of the original seismic process, we needed to avoid procedures like cleaning, filtering or declustering. Otherwise it would be impossible to preserve original time structure of earthquakes distribution. This, together with the necessity to have as possible long data sets, forced us to select as possible long period of observation with as possible low representative threshold. Such compromise, when catalogue is long enough and completeness threshold is as low as possible, according to results of time completeness analysis, seemed to be possible from 1975. Indeed, in Fig. 3, we see that since the middle of 70th of the last century Mc clearly decreased, what finally enabled us to work with southern Californian earthquake catalogue with magnitude of completeness M=2.6, according to the Gutenberg–Richter relationship analysis (see Fig. 2). We understand that in such catalogue we deal with both independent, as well as dependent (aftershocks or foreshocks) events, but in the frame of aims, targeted in the present work this is quite acceptable, because we speak about general temporal behavior of seismic process and because, as it is known, physics of generation of dependent and independent events is similar (See e.g. [Davidsen, Goltz, Geophys. Res. Lett.31(2004), pp. L21612.; P. Bak, C. Tang, K. Wiesenfeld, Phys. Rev. A 38(1) (1988), pp.364–374]).

[Figure]

Fig. 1.

[Figure]

Fig. 2.

[Figure]

Fig. 3.

We would like further underline that, in any case to assess the possible influence of dependent events on the results of our calculations, we performed analysis at higher representative thresholds M3.6, M4.6 and even for M5.6. According to our analysis dependent events do not essentially influence results of IDT analysis.

Reviewer is correct saying that author of manuscript "do not make distinctions between depths of hypocenter". From above said it should not be surprising that we do not wanted to differentiate entire process by hypocenters depths and thus change the time structure of original earthquake occurrences. As we pointed above, from the same logic we do not make any catalogue cleaning, declustering, etc. Again, this was quite logical for the targeted research purpose, aiming at the analysis of temporal features of the original (natural) seismic process. This goal to be correctly achieved necessitates avoiding artificial distortion of original dynamical features of earthquakes time distribution, what usually is impossible by any cleaning or filtering of catalogue (especially of such high quality as used in our work south Californian earthquake catalogue). We base our analysis on the often practice, when [see e.g. P. Bak, in (How Nature Works: The Science of Self-Organized Criticality, 1996); Christensen et al.(in Proc.Natl. Acad. Sci. U.S.A. 99, 2509, 2002); Corral (in Phys.Rev.Letters, 2004); Corral (in Phys. Rev. E 68, 035102(R) 2003); etc.] seismic processes in catalogue is regarded as a whole, irrespective of the details of tectonic features, earthquakes location or their classification as mainshocks or aftershocks. Thus, we logically abandoned also differentiation of earthquakes according to depths of hypocenters.

In fact, answers to the almost all questions of reviewer 2, are already done in one of the famous articles of Alvaro Corral (in Phys. Rev. E 68, 035102(R) 2003) where it is said that view similar to used in our analysis ".. follows one of the key guidelines of complexity philosophy, which is to find descriptions on a general level; the existence of general laws fulfilled by all the earthquakes unveil a degree of unity in an extremely complex phenomenon".

The authors
don't even identify the spatial region, they simply took the data in the archive taken
without criticism. They don't select the main shock from aftershocks.

Answers to these remarks, see above.

So the statistical
description and the results are affected by these undefined choices.

Here we completely agree with the statement of reviewer 2. Indeed, our results obtained by analyzes accomplished by the carefully tested IDT method, express features of earthquakes' time distribution in the original catalogue, in which the temporal structure of seismic process (as possible) is not distorted by the some, not always well grounded, procedures. Unfortunately, blind inclinations of some researchers to change reality in accordance with their personal preferences or to make "defined choices", especially when we deal with complex process, often lead to unscientific and incorrect conclusions. Thus, YES, we agree that results really are affected by the features of natural earthquake's time distribution. Moreover, our results reflect features of this natural (as possible untouched) seismic process. This is why they are new and important, as they show changing in time extent of regularity and periods, when seismic process is most random-like.

The Conclusions are trivial.

We would sincerely appreciate reviewer 2, if he/she could provide in depth explanation why our results can be regarded as trivial. In the report of reviewer 2, we do not see any documentation indicating that our findings are something well known or not deserving any attention. Especially we'd be glad to get references, in which it is shown convincingly that the extent of randomness in earthquake time distribution is changing over time and that there are better methods applicable to short periods, when the seismic process is closer to randomness.

CONCLUDING REMARK The goals of the work are not well motivated; it seems to be
a mere statistical exercise.

It is said in manuscript that the motivation for the present work was to assess how the extent of regularity in the earthquakes time distribution changes over the considered period of catalogue time span. This problem, in spite of wide scientific interest [e.g. Davidsen, C. Goltz, 2004; Kawamura. 2007; Kenner, M. Simons, 2005; etc.] still remains unanswered. At the same time, it is clear that without such knowledge the better understanding of seismic processes can not be achieved. Moreover, scientific posing of such general tasks as earthquake prediction or forecast, will not look well-grounded unless basic features of seismic process dynamics in spatial, temporal or energy domains will not be understood.

We think that analysis of simulated data sets, carefully accomplished in our research, should not be regarded just as "statistical exercise". The matter is that one needs to fulfill analysis by suggested IDT method on simulated complex data sets with (predefined) different extent of randomness in order to apply the method to seismic data sets with unknown complex structure. Only after such comparative analysis (calibration) and appropriate data selection we could undoubtedly prove that IDT approach is able to discern and quantify the changes in the complexity level of the process even in the case when we deal with short data sets from a complex process like seismicity. So, this analysis was a necessary part of research aimed to present and launch the new method of IDT.

Besides, in our opinion results obtained from the careful analysis of different simulated random data sets, given compactly in one article, will be undoubtedly helpful for researchers from different fields for different testing purposes.

So the paper needs a deep afterthought.

We corrected our manuscript significantly. The revised version of manuscript contains a result of additional work and testing of data sets.

---

## Short Comment (SC1) · 15 Mar 2018

Revised manuscript sent 14 Mar 2018 should be deleted. Corrected version was uploaded today

---

## Author Comment (AC6) · 15 Mar 2018

The comment was uploaded in the form of a supplement:
https://www.nonlin-processes-geophys-discuss.net/npg-2017-77/npg-2017-77-AC6-supplement.pdf

---

## Author Comment (AC7) · 19 Mar 2018

The comment was uploaded in the form of a supplement:
https://www.nonlin-processes-geophys-discuss.net/npg-2017-77/npg-2017-77-AC7-supplement.zip

---

## Author Comment (AC8) · 19 Mar 2018

The comment was uploaded in the form of a supplement:
https://www.nonlin-processes-geophys-discuss.net/npg-2017-77/npg-2017-77-AC8-supplement.zip

---

## Author Comment (AC9) · 19 Mar 2018

Dear Editor, Hereby we send revised version of our manuscript npg-2017-77. This version of manuscript was prepared according to NPG standards. All comments of referees have been answered in the revised text or in the separate text of answers to referees. All changes and additions, made according to referees suggestions, are given in red in the marked-up manuscript version. Figures 2-13 and corresponding legends have been changed or updated. Sincerely, T. Matcharashvili

P.S. Changed are marked by red color in the manuscript.
* * *

---

## Author Response (AR2)

Submitted on 19 Apr 2018
Anonymous Referee #1

First of all let me again express my sincere gratitude to reviewer 1, for fair and highly professional reviewing of our manuscript. With no regard of final decision, I appreciate importance of most of his/her remarks in the improving of our manuscript.

Below are answers to the reviewer's remarks.

Reviewer 1. Even though the authors took large effort to respond to all comments, they could not dispel my major concern about the appropriateness of the introduced IDT measure to describe the degree of randomness. The complete paper is based on the assumption that for random sequences sum_[i=1..N] DT(i) → 0, for large N. This needs to be proofed.
Referring to a paper in preparation or stating " logically, for any random sequence, the sum of the deviation times should approach zero, when " (revised version, page 2, last paragraph) is not sufficient. The sentence needs mathematical background. I assume the authors have something like the law of large number in mind, but this is not valid in their application, since it requires independent identical distributed data. Yet, DT(i) is not independent from DT(i-1).

*I respect opinion of reviewer 1, but as it was pointed many times we base present analysis on the strong empirical argumentation of used approach. This do not exclude possibility that we come back to theoretical proofs in future but here we believe that our work should to be evaluated as it is, i.e. from the point of view of correctness of presented empirical results. Again, in this work we aimed to present idea how interesting dynamical features of complex data sets can be discerned through the simple method, effectiveness of which is clearly demonstrated empirically for both modeled as well as real data sets from original seismic catalogue.*

Reviewer 1. Also the presentation of a matching example (as the analysis of colored noise) is not sufficient to proof the concept. In fact, Fig. 5a) contradicts the authors assumption, since IDTs for shorter time series (e.g. N = 5000) are smaller than IDTs for longer time series (N=34020) and consequently do not converge to zero for large N.

*We agree with reviewer that former version of Fig.5 is better to be corrected. Apparently, it is better to show results normed not only to the span of window (i.e. sum of data in window) but also to the number of data in each window. In this case it is easier to understand that IDT for longer windows goes closer to zero. Thus, results in present Fig.5 do not contradict to our working assumption.*

Reviewer 1. Moreover, a deterministic time-series with equidistant time steps contradicts the authors assumption that the IDT for more random-like time series is closer to zero, than for deterministic time series (even though a equidistant time series is not in the interest of the study). I could think of several other deterministic time series, where this is the case.

*We have already explained that comparison with equidistantly distributed markers has no sense in our analysis. Thus, we clearly stated that IDT analysis in its present form should be used for non equidistantly distributed marker sequences. The question why we need to avoid equidistant marker sequences is not easy to answer and is related with the more general problem of what really random process is and what is relation between randomness and order. In other words it is*

*equal if you pose the question – could we regard order as a limit of randomness or not? I can not remember sources which directly answer such fundamental questions. Moreover I do not think that this article about the simple empirical analysis really necessitate to be expanded to such fundamental discussions. At the same time, I agree that problem is of immense general importance and hope will be able come back to this question later. As for reviewers remark on deterministic time series, I think here we face misunderstanding. In general, each time series of any kind is equidistant by the definition (otherwise it would be unevenly sampled sequence). From this point of view equidistant sequence, what we mean in manuscript, is rather time series consisted of equal values. At the same time is not clear why deterministic processes should produce data sets of this kind? From thermodynamics point of view this will be the condition of thermodynamic equilibrium for which the term "deterministic" is questionable, at least in the present connotation of deterministic processes.*

Reviewer 1. revised paper, page 7: "From this figure [fig. 6] we see that IDT values goes closer to zero when the extent of order decreases. Besides, it also becomes clear that even for short data sets IDT calculation is useful to detect differences in considered data sets."
Figure 6 only shows, that the distributions for low orders are more narrow than for higher orders. Yet, the mean seems to be the same for all distributions and differs from zero (~0.8?). Since sequences of high order magnitude also have good chances to be close to zero, the usefulness of IDT to detect differences is questionable. Several changes will be undetected.
On top, neither Figure 6 nor any other analyses shows, how IDT reacts to changes in the regularity of the time series. For all synthetic data sets the spectral exponent stays constant.

*Here we, again, apparently face misunderstanding of the meaning of curves in Fig.6. In manuscript we speak that "IDT values goes closer to zero when the extent of order decreases". Indeed, for the case $\beta$=0.001(triangles) corresponding curve crosses ordinate axis at about 45 % while for$\beta$=1.655(cross signs) at about 15%. How this fact can be interpreted otherwise if not the indication that at lower extent of order ($\beta$=0.001) larger part of IDT values is closer to zero (at least three times) comparing to more regular sequences (other than $\beta$=0.001)? This is why, at the same length of windows, distribution becomes narrower for less ordered sequence. It should not be forgotten that we deal with finite length (even very short) data sets (windows) for which closeness to zero, of IDT values, can be regarded just statistically. I can not agree with reviewer and would like to underline again that results in Figures 5 and 6 exactly show that changes in the extent of regularity of used synthetic data sets, lead to decrease IDTs closer to zero value. Spectral exponents, beta values, are already shown in figures.*

Reviewer 1. Revised paper, page 9: "Increasing the threshold to M3.6, M4.6, and M5.6 leads to following IDT values: 71.7, 6.7, -0.87 accordingly. Two important things can be underlined here: first, the increase of the magnitude threshold makes the time distribution of remained EQs more random and second, according to our conjecture to the more random EQ distribution should corresponds the closer to zero IDT value, what indeed is shown above. "
The IDT values depend on the number of events in the time series. Consequently the IDTs for the different threshold magnitudes are not comparable and can not be used to support the given conclusions.

*I am very grateful to reviewer 1, for this very important remark and completely agree that "IDTs for the different threshold magnitudes are not comparable". I regret that we normed IDT values only to the time span, while it was necessary to norm also to the number of events (in that cases when number of events in windows have been different). Now we add norming to number of data*

*and see that stronger events do not differ too much from smaller ones by their time distribution features.*

Reviewer 1. Revised paper, page 10: "Thus, comparing the average of integral deviation times, calculated for the entire length of randomized catalogues, with the IDT value of the original SC catalogue, we see that the last one is two orders of magnitude larger."
Comparing the average IDT of randomized catalogues to the original catalogue is not meaningful comparison, since the average value can be expected to be closer to zero in any case. It would be reasonable to instead show the IDT of the original catalogue in Fig 7, which gives the distribution of IDTs of randomized time series.

*Here I can not agree with reviewer 1, though can not exclude that he misunderstood what was done with randomized catalogues. So let me once again explain that for testing purpose we compiled randomized catalogues in which the time occurrences of earthquakes from original catalogue, were randomly shuffled. For each of such catalogs we calculated IDT values and normed to the time span (norming to the number of EQs was not necessary because was the same as in the original catalogue). Than we compared all these 150 IDT values ( -7.2 ± 57.5) with original one (-659.15). Z score =11.2, corresponding to p=0.001 convince that each IDT from randomized catalogues is significantly smaller than for original. Figure below will additionally help to understand situation that IDT values from each of random catalogues is essentially smaller than ID from original catalogue and thus effects of averaging cannot play any role.*

[Figure]

Reviewer 1. Revised paper, page 11: "the strongest earthquakes never occurred in periods, when IDT curve comes close to zero value or crosses abscissa line" …
This is not very surprising, since only 6 of the 3500 time windows correspond to the large magnitude earthquakes. It is quite unlikely to hit one those time windows.

*Can not agree with Reviewer's statement. In response I could say that the probability that certain earthquake will hit the certain time window is unlikely, exactly in the same way. Yes in Fig. 8b, we are focused on 6 windows (with main earthquakes far from zero crossings) out of 3500, but it is noticeably that all 6 windows are located at the rising branches of IDT curve (this is indeed unlikely to be happened by chance) and usually are very far from zero crossings (even first strong event M6.4) occurred after 100 events from crossing. All these, i.e. deviations from zero crossings and location on rising branches can not be explained as something caused by blind chance, though definitely needs further deep analysis, what we plan to do in nearest future.*

Reviewer 1. Revised paper, page 12: "As we see in Fig. 9, analysis carried out on shorter catalogues, confirm the result obtained for the entire period of observation (1975-2017) and convinces that the curve of IDT values crosses abscissa at periods of relatively decreased seismic energy release.".
Still it is unlikely to hit one of the high magnitude EQ windows. Fig. 9 also shows that the crossing of the abscissa strongly depends on the starting point of the studied period (e.g. the points for 1980 differ strongly compared to 1985). Consequently, I doubt there is a meaningful interpretation of the crossing points.

*In order discussion to look more scientific let we speak about testable facts but not about what is likely or not. Exactly, our analysis clearly shows that windows with high magnitude EQs almost never coincides with windows with closest to zero IDT values. In present version we changed figures 9, 10 and 11 and added tables where are indicated windows with strong earthquakes and windows when IDT is closer to zero. From table 1 we see that for threshold value M2.6 no strong earthquakes (M6.4 -M7.3) occurred in windows with IDT closest to zero. In tables 2 and 3, for M3.6 and M4.6 representative thresholds we generally observe the same. Only in one case we found that strong earthquakes occurred in windows with closest to zero IDT values. This may be somehow is caused by specificity of seismic process in considered time periods and apparently do not in principle contradict the supposition that strong earthquakes rarely occurred in periods when calculated IDT value is closer to zero.*

Reviewer 1. In general, it is difficult to compare the IDT values you provide, since some are not normalized, some are normalized to the span of window (e.g. Fig. 5), some to number of events (e.g. Fig 9). A uniform presentation should be used. E.g. it is difficult to compare the IDT of the original SC catalogue (not normalized) with the distribution of IDTs for randomized catalogues (normalized to span of window).

*In present version in all cases we show results normalized to number of events and span of windows. Exclusion are figures 12 and 13 where we compare results in similar 100 data length windows so here normalization to the number of events was not necessary.*

Reviewer 1. The results of Fig. 5b do not match with Fig. 6. The mean values for high orders are larger in Fig 5b compared to the means in Fig. 6.

*This is because in Fig.6 we present PDFs, not frequencies.*

Reviewer 1. I still don't understand the authors' concept of randomized time series, since a shuffling of occurrence times will not change anything. Maybe the authors mean a shuffling of waiting times.

*As it was already explained we consider randomized catalogue in which time locations of earthquakes from the original catalogue have been randomly changed.*

Submitted on 31 Jan 2018
Anonymous Referee #2

In opposite to remarkable work of reviewer 1, I am deeply disappointed by human and scientific irresponsibility of reviewer 2.

Indeed, we have answered all his/her remarks in revision 1, and seems he/she has no further arguments against.

In spite of this fact, reviewer 2 recommends rejection without any arguments. In the same way, not providing any arguments, I also can state that reviewer2 is incompetent in the subject of our manuscript and do not deserves to be reviewer of respected NPG.